# Evaluating different machine learning methods to simulate runoff from extensive green roofs

Elhadi Mohsen Hassan Abdalla[1], Vincent Pons[1], Virginia Stovin[2], Simon De-Ville[2], Elizabeth Fassman-Beck[3], Knut Alfredsen[1], and Tone Merete Muthanna[1]

[1]Department of Civil and Environmental Engineering, The Norwegian University of Science and Technology, Trondheim, 7031, Norway
[2]Department of Civil and Structural Engineering, The University of Sheffield, Sheffield, S1 3JD, United Kingdom
[3]Southern California Coastal Water Research Project, Costa Mesa, California, CA 92626, United States

**Correspondence:** Elhadi Mohsen Hassan Abdalla (Elhadi.m.h.abdalla@ntnu.no)

**Abstract.** Green roofs are increasingly popular measures to permanently reduce or delay stormwater runoff. The main objective of the study was to examine the potential of using machine learning (ML) to simulate runoff from green roofs to estimate their hydrological performance. Four machine learning methods, Artificial Neural Network (ANN), M5 Model tree, Long Short-Term Memory (LSTM) and k-Nearest Neighbour (kNN) were applied to simulate stormwater runoff from sixteen extensive green roofs located in four Norwegian cities across different climatic zones. The potential of these ML methods for estimating green roof retention was assessed by comparing their simulations with a proven conceptual retention model. Furthermore, the transferability of ML models between the different green roofs in the study was tested to investigate the potential of using ML models as a tool for planning and design purposes. The ML models yielded low volumetric errors that were comparable with the conceptual retention models, which indicates good performance in estimating annual retention. The ML models yielded satisfactory modelling results (NSE > 0.5) in most of the roofs, which indicates an ability to estimate green roof detention. The variations in ML models' performance between the cities was larger than between the different configurations, which was attributed to the different climatic characteristics between the four cities. Transferred ML models between cities with similar rainfall events characteristics (Bergen-Sandnes, Trondheim-Oslo) could yield satisfactory modelling performance (NSE>0.5, |PBIAS|<25%) in most cases. However, we recommend the use of the conceptual retention model over the transferred ML models, to estimate the retention of new green roofs, as it gives more accurate volume estimates. Follow-up studies are needed to explore the potential of ML models in estimating detention from higher temporal resolution datasets.

## 1 Introduction

Green roofs are a type of green infrastructure (GI) that have received significant attention in recent years. In contrast to conventional stormwater infrastructure, green roofs attempt to decrease stormwater outflows while providing other services, such as reducing urban heat island effect, preserving the cities ecosystems and improving the urban visual amenity among other benefits (Berndtsson, 2010) . Roof areas represent around 40-50% of impermeable areas in dense urban catchments (Dunnett and Kingsbury, 2004); therefore, retrofitting current roofs with substrate/growing media and vegetation offers an

efficient and area-free GI option. Many studies have confirmed the potential of green roofs to mitigate rainfall events from field measurements (Fassman-Beck et al., 2013; Johannessen et al., 2018; Liu and Chui, 2019; Stovin, 2010).

Quantifying the hydrological performance of a green roof is usually done by estimating **retention**, a permanent reduction of stormwater by evapotranspiration, and **detention**, flow peak reduction and delay. Both retention and detention metrics are needed to justify the widespread implementation of green roofs by the stormwater community, and for planning and design by practicing engineers. Hence, numerous studies have investigated different approaches and tools to simulate outflows from green roofs to estimate retention and detention metrics.

For estimating green roof detention, models that simulate rainfall-runoff events in short time steps (sub-hourly) are required. Several models have been tested successfully in the literature, which can be categorized into physically-based and conceptual models. Physically-based models simulate the water flow in porous media by solving physical equations numerically, such as the Richards equations, either in 1D (Bouzouidja et al., 2018; Liu and Fassman-Beck, 2017; Peng et al., 2019), 2D (Li and Babcock Jr, 2015; Palla et al., 2009) or 3D (Brunetti et al., 2016). Several tools exist that can be used to implement this type of 35    models, such as HYDRUS (Simunek et al., 2005), SWMS-2D (Simunek et al., 1994) and Comsol multiphysics (Multiphysics, 2013; Sims et al., 2019). These models have proven to be accurate and to rely only on measurable parameters (Sims et al., 2019) and can be powerful tools for studies that aim at in-depth understanding of the hydraulic behaviours of the different green roof layers (Brunetti et al., 2016).

Another category of physically-based models apply simplified and analytical forms of physical equations, such as the Green-40    Ampt equation for infiltration and Darcy law for saturated water flow (Krebs et al., 2016; She and Pang, 2010; Hernes et al., 2020). Popular modelling tools that implement these models include the EPA-SWMM (Rossman et al., 2010) and Mike-Urban (DHI, 2017). This category of models is perhaps the most commonly applied in the literature of green roof modelling, and it has been acknowledged by many studies to be a suitable tool for analysing the hydrological performance of green roofs (Cipolla et al., 2016). However, due to the simplicity of these models, they often rely on calibrated rather than measured parameters.Peng 45    and Stovin (2017) found the simulated hydrographs of uncalibrated SWMM models to deviate significantly form the observed ones. Johannessen et al. (2019) attempted to transfer calibrated SWMM model parameters between similar green roofs located in different locations. However, only parameter sets from wet locations yielded good results in drier locations but not vice versa

Conceptual models simplify the physical processes using linear or nonlinear equations to simulate green roof runoff. One common type of these models is the reservoir routing model which was applied to estimate runoff detention from event-based 50    simulations in previous literature (Palla et al., 2012; Soulis et al., 2017; Vesuviano et al., 2014). These models were found to produce results that are comparable to physically-based models with lower level of complexity (i.e. reduced number of model parameters) (Peng et al., 2019). Palla et al. (2012) recommended the use of a reservoir routing model instead of physically-based models for design purposes when little information is available about the green roof properties. However, the parameters of conceptual models are not measurable. Hence, calibration is needed to find their optimal values, unlike physically-based 55    models (Peng et al., 2019). A few studies have identified relations between the flow parameters of reservoir models and some physical properties of green roofs, such as slope and substrate depth (Vesuviano and Stovin, 2013; Yio et al., 2013). However, these studies focused on lab-scale green roofs in which detention due to the horizontal flow is not significant (Sims et al., 2019).

For estimating green roof retention, models with water balance equations (in hourly or daily time step) and suitable representation of the actual evapotranspiration process (AET) were found by many studies to be sufficient (Bengtsson et al., 2005; Jahanfar et al., 2018; Johannessen et al., 2017; Stovin et al., 2013). The most common way to model AET is by multiplying the potential evapotranspiration (PET), the maximum evaporation rate assuming unlimited water supply, with reduction functions that account for soil moisture deficit and crop type. The reduction functions require careful parameterization of the maximum storage of the roof and crop factors. The maximum storage of the roof was found by many studies to be related to the measurable field capacity of the substrate (Liu and Fassman-Beck, 2017; Stovin et al., 2013). Crop factors for agricultural crops are well documented and studied (Allen et al., 1998). However, crop factor values for Sedum plants, commonly applied for green roofs, are less known. Previous studies reported different crop factor values for Sedum plants (Berretta et al., 2014; Rezaei et al., 2005; Sherrard Jr and Jacobs, 2012).

Data-driven models, which are derived entirely from observed data, may offer alternative modelling tools that can estimate both retention and detention of green roofs without explicitly accounting for complex hydrological processes. However, the use of data-driven models in green roof studies has been limited to simple regression models (Carson et al., 2013) which are site-specific and not transferable. More advanced data-driven methods, such as Machine learning (ML), have been commonly applied in many hydrological modelling studies in the last few decades. However, only a few studies were found to apply ML models in green infrastructures (Tsang and Jim, 2016; Radfar and Rockaway, 2016; Li et al., 2019) and no study was found to apply ML models in estimating the hydrological performance of extensive green roofs.

Machine learning methods have been successfully applied in hydrological modelling in recent decades. Previous studies reported better performances of ML models compared to conventional hydrological models in runoff prediction (Solomatine and Dulal, 2003; Yilmaz and Muttil, 2014; Young et al., 2017), runoff simulation (Javan et al., 2015; Kratzert et al., 2018) , and for building relationships between water level and discharge (Bhattacharya and Solomatine, 2005). Some of the popular Machine Learning methods applied in hydrology include Artificial Neural Networks (ANN), M5 model tree, Long Short-Term Memory (LSTM), and k Nearest Neighbours (kNN).

Artificial Neural Network is the most common and among the earliest ML used in hydrological modelling (Daniell, 1991). Early examples of research into ANN includes the study conducted by Hsu et al. (1995), in which ANN outperformed the linear ARMAX and the conceptual Sacramento SAC-SMA model in simulating runoff from a medium-sized catchment. Likewise, Tokar and Johnson (1999) compared an ANN to a simple conceptual model and found the former to outperform the latter. M5 model tree has been applied in different studies. Solomatine and Dulal (2003), reported a satisfactory performance of both M5 model tree and ANN in runoff forecasting. They, however, emphasized the advantages of M5 model tree over ANN due to the better interpretation of M5 model outputs. Goyal et al. (2013b) applied the M5 model tree for flow forecasting in India, among other ML methods and found it to perform satisfactorily. Away from flow simulation, Gharaei-Manesh et al. (2016) used M5 tree and other methods to simulate the spatial distribution of snow depths in Iran, while Goyal et al. (2013a) evaluated M5 model tree on formulating operation rules for a reservoir. Kisi (2016) used M5 model tree to model reference evapotranspiration.

LSTM has been applied in different scientific fields and could provide good results (Shen, 2018). Regarding runoff modelling, Kratzert et al. (2018) investigated the potential of LSTM to predict runoff from ungauged basins. They could achieve good prediction performance that was comparable to the well-known Sacramento model. Similarly,Ayzel (2019) obtained comparable results with LSTM to a conceptual model. Hu et al. (2018) compared between an ANN and an LSTM in runoff simulation and found the latter to outperform the former. Nevertheless, LSTM is computationally expensive, and the training process takes a long time (Ayzel, 2019). k-Nearest Neighbour was applied first by Karlsson and Yakowitz (1987) in runoff forecasting in which it outperformed unit hydrograph forecasters. Modaresi et al. (2018) found the k-Nearest Neighbour to be comparable with ANN in monthly runoff forecasting. Furthermore, Wu et al. (2009) applied the k-Nearest Neighbour in predicting monthly runoff, and they discussed the effect of k value on the performance of kNN.

Few studies have modelled green infrastructure with ML techniques. For instance, Tsang and Jim (2016) applied a Fuzzy-neural network to optimize irrigation of a green roof by estimating soil moisture deficit. The neural network could reproduce the soil moisture well, which indicates the capability of ML models to simulate the nonlinear AET process. Li et al. (2019) developed an artificial neural network model to predict the flow reduction from a catchment with different GI structures. Similarly, Radfar and Rockaway (2016) applied a neural network model to predict flow reduction from a permeable pavement. The satisfactory performances of ML models in two studies demonstrate the potential of ML models in GI hydrological modelling.

This study examines the ability of four machine learning methods, M5 model Tree, Artificial Neural Networks (ANN), Long Short-Term Memory (LSTM), and k-Nearest Neighbour (kNN), to estimate green roof hydrological performance, specifically by:

1. Evaluating the performance of ML models in simulating the temporal dynamics of green roof subsurface runoff and estimating the retention from long term simulations across different climatic locations.

2. Investigating the potential of using ML models as a useful tool for planning that predicts the performance of new green roofs when observations are not available.

## 2 Data

Sixteen extensive green roofs located in four Norwegian cities with different climates: Bergen (BERG), Sandnes (SAN), Oslo (OSL) and Trondheim (TRD) were used in the study. Bergen city is located on the western coast of Norway. Bergen is the wettest city among the four with annual precipitation of 3110 mm followed by Sandnes city, which is located on the south-west coast, with annual precipitation of 1690 mm. Oslo is the driest city with only 970 mm of annual precipitation while Trondheim is the northmost city with annual precipitation of 1070 mm. According to the Köppen–Geiger Climate Classification (Kottek et al., 2006) , both Bergen and Sandnes are classified as temperate oceanic climate (Cfb), while Oslo has the warm-summer humid continental climate (Dfb) and Trondheim has a subpolar oceanic climate (Dfc). The locations of the four cities are shown in figure 1. Table 1 shows the geometries and configurations of roofs. Roof geometries (areas and slopes) vary between

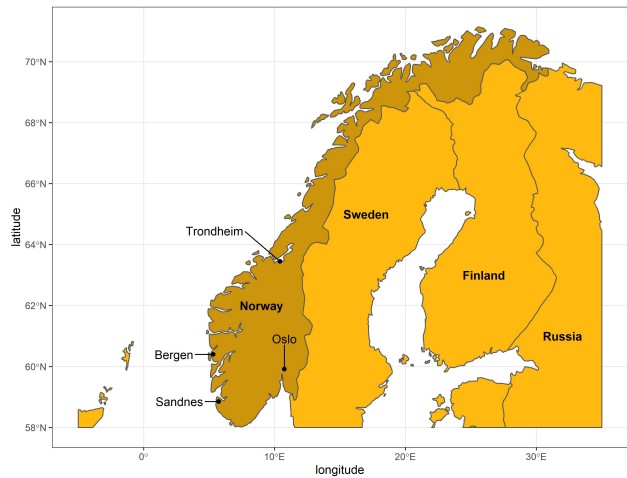

**Figure 1.** Locations of the four Norwegian cities with green roof field data

the cities, while the different configurations represent the variety of options in the Norwegian green roof market. Some green
roofs in the study have the same configuration, for instance, BERG1, SAN1 OSL3 and TRD1. Continuous times series data
were collected from TRD, BERG and SAN roofs between 2015 to 2017, while the green roofs at OSL have a seven-year record
of data from 2011 to 2017. Data includes precipitation, runoff, relative humidity and wind speed at a 1 min resolution. In Oslo,

the wind speed was not measured at the roofs but collected from a nearby station. For details about roof setup, data collection and processing, please refer to Johannessen et al. (2018).

**Table 1.** Roof Geometries and Configurations

| Roof | Geometry | | | Configuration | | | |
|------|-------|--------|-------|----------------------------|-------------------------------|---------------------------------------------------------------|---------------------|
| | Width | Length | Slope | Vegetation mat thickness[1] | Extra Substrate type and thickness | Drain mat type and thickness | Total roof thickness |
| | (m) | (m) | (%) | (mm) | | | (mm) |
| BERG1 | 1.6 | 4.9 | 16 | 30 | - | Textile retention fabric (10mm) | 40 |
| BERG2 | 1.6 | 4.9 | 16 | 30 | - | Substrate mat[2] (50mm) | 80 |
| BERG3 | 1.6 | 4.9 | 16 | 30 | Separate Substrate[3] (50mm) | Drainage layer[5] (EPS) (75mm) + Textile retention fabric (5mm) | 160 |
| BERG4 | 1.6 | 4.9 | 16 | 30 | - | Textile retention fabric (3mm) | 33 |
| BERG5 | 1.6 | 4.9 | 16 | 30 | Pumice (50mm) | Textile retention fabric (3mm) | 83 |
| OSL1 | 2 | 4 | 5.5 | 30 | - | Drainage layer (HDPE)[6] (25mm) | 55 |
| OSL2 | 2 | 4 | 5.5 | 30 | Separate Substrate[3] (50mm) | Drainage layer (HDPE)[6] (40mm) + Textile retention fabric (5mm) | 125 |
| OSL3 | 2 | 4 | 5.5 | 30 | - | Textile retention fabric (10mm) | 40 |
| SAN1 | 1.6 | 5.3 | 27 | 30 | - | Textile retention fabric (10mm) | 40 |
| SAN2 | 1.6 | 5.3 | 27 | 30 | Separate Substrate[3] (50mm) | Drainage layer (EPS)[5] (75mm) + Textile retention fabric (5mm) | 160 |
| SAN3 | 1.6 | 5.3 | 27 | 30 | - | Textile retention fabric (3mm) | 33 |
| SAN4 | 1.6 | 5.3 | 27 | 30 | - | Substrate mat[2] (50mm) | 80 |
| TRD1 | 2 | 7.5 | 16 | 30 | - | Textile retention fabric (10mm) | 40 |
| TRD2 | 2 | 7.5 | 16 | 30 | - | Substrate mat[2] (50mm) | 80 |
| TRD3 | 2 | 7.5 | 16 | 30 | Separate Substrate[3] (50mm) | Drainage layer (HDPE)[6] (25mm) + Textile retention fabric (5mm) | 110 |
| TRD4 | 2 | 7.5 | 16 | 30 | Separate Substrate[3] (50mm) | Drainage layer (PE)[4] (30mm) | 110 |

[1] Pre-grown reinforced vegetation mats (sedum)

[2] Substrate mat: a mineral wool plate

[3] Separate Substrate: a mixture of Leca and bricks

[4] Drainage layer (PE): plastic drainage layers of polyethylene

[5] Drainage layer (EPS): plastic drainage layers of expanded polystyrene

[6] Drainage layer (HDPE): plastic drainage layers of high-density polyethylene

## 3 Methods and tools

### 3.1 Machine learning models

#### 3.1.1 M5 model Tree

In this approach, the training data are divided into many subsets. For each subset, a piece-wise linear regression equation is
built between the output and the input variables (Solomatine and Dulal, 2003). The algorithm used by the model tree is called
M5, which was developed in 1992 (Quinlan et al., 1992). It divides the data into subsets based on rules that reduce the intra-
variation (variance) within each subset (variables within each subset are as similar as possible). The M5 model tree has an
upside-down tree structure. Input variables enter the tree from the top (the tree root) to arrive at the models located at the tree
leaves. For a detailed explanation of the M5 model tree, see Solomatine and Dulal (2003).
In this study, Cubist library in R (Kuhn et al., 2012) was used to build M5 models. The performance of Cubist-M5 models
can be improved by tuning two hyperparameters, namely *committees* and *neighbours*. The former is the number of trees in a
boost-like ensembles scheme where iterative M5 models trees are built in sequence. The first M5 tree is built following the M5
algorithm, while the subsequent trees are created from the residuals of the single tree. The final model prediction is the average
from all M5 trees in the ensemble. The final prediction of a single tree can be improved by a post-model nearest-neighbour
adjustment (Quinlan, 1993). The predicted value of the tree is smoothed following a weighting schemes from several nodes
within the single tree. The number of nodes used in the smoothing is called *neighbours*.

#### 3.1.2 Artificial Neural Network (ANN)

The ANN applied in this study is the standard feed-forward neural network. It comprises an input layer, a hidden layer(s) and
an output layer. The building block of the network is called a neuron, and each neuron is fully connected with all other neurons
in the backward and forward layers. Hidden layers are where relations between input variables are revealed. Each neuron in
the ANN applies simple mathematical operations for the variable vectors, as represented in equation 1:

$$O = f(X1 \times W1 + X2 \times W2 + B) \tag{1}$$

$O$ is the output from a neuron, $W1$ and $W2$ are the weights of the variables $X1$ and $X2$, respectively, and $B$ is the neuron's
bias. $f(.)$ is the neuron's activation function that adds non-linearity to the neuron's output. During the training process, the
weights and biases are updated for the whole network to obtain the best fit between simulated and observed outputs. A standard
algorithm used for the training is backpropagation, which uses the approach of the steepest gradient descent (Rumelhart et al.,
1986). Training of the neural network is done by dividing the training data set into several batches. The weights and biases
are updated for each batch until all training data have been visited, and then the same cycle is repeated. This cycle is called
an epoch, and the learning performance improves with the increasing number of epochs. However, there is a risk of overfitting
for models with high numbers of epochs. To avoid that, a separate data set (validation data set) is often used to optimize the

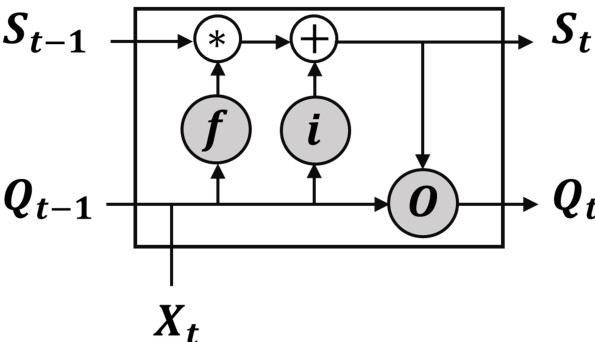

**Figure 2.** : Structure of the Long Short-Term Memory (LSTM) cell, modified from Kratzert et al. (2018)

number of epochs by determining the objective function for the validation data while training the model. Overfitting starts when the error increases in the validation data set while decreasing in the training data.

### 3.1.3 Long Short-Term Memory (LSTM)

In hydrology, sequential runoff data are often autocorrelated, especially data with a short time step. Autocorrelation is triggered by system memory in hydrology, usually due to the storage effects. A Recurrent Neural Network (RNN) is a special type of neural networks that can tackle sequential data modelling because it includes output from the previous time step as input to the following time step. Nevertheless, it doesn't account for the long-term dependency in the system. Hochreiter and Schmidhuber (1997) discussed the issue of RNN with long term dependency and proposed a unique RNN model called Long Short-Term Memory (LSTM). In this model, a value representing the system memory $(S)$ is calculated and updated each time step to account for the long-term dependency of the system. LSTM cell comprises of three gates (Figure 2): forget gate $(f)$, input gate $(i)$ and output gate $(o)$. The three gates control cell output and update its state for each time step by applying weights $(W)$ and biases $(B)$. The first step is to control which information to be forgotten from the previous time step $(f_t)$, which is done by the forget gate using equation 2. Secondly, the updated value for the cell state $(\Delta S_t)$ is determined from equation 3. Subsequently, the input gate output $(i_t)$ is derived from equation 4, which controls how much information will be used from $\Delta S_t$ to update the cell state $S_t$. In the following step, the cell state $S_t$ is determined by applying equation 5. Finally, the output from the output gate $(O_t)$ is calculated from equation 6 which used to determine the cell output $(Q_t)$ by using equation 7. In this study, Keras library (Chollet et al., 2015) was used to build ANN and LSTM models.

$$f_t = f_f(W_f \times X_t + U_f \times Q_{t-1} + B_f) \tag{2}$$

$$\Delta S_t = f_{\Delta S_t}(W_{\Delta S_t} \times X_t + U_{\Delta S_t} \times Q_{t-1} + B_{\Delta S_t}) \tag{3}$$

$$i_t = f_i(W_i \times X_t + U_i \times Q_{t-1} + B_i) \tag{4}$$

$$S_t = f_t \cdot S_{t-1} + i_t \cdot \Delta S_t \tag{5}$$

$$o_t = f_o(W_o \times X_t + U_o \times Q_{t-1} + B_o) \tag{6}$$

$$Q_t = \tanh S_t \cdot o_t \tag{7}$$

### 3.1.4 k Nearest Neighbours (kNN)

k Nearest Neighbours is a nonparametric method that estimates the output of each time step based on its similarity to the historical time steps. Basically, the algorithm determines similarity distances between each of the input variables of the new time step to the variables from the training data set. Then it calculates the mean outputs of the k most similar time steps.In this study, the euclidean distance was used as a measure of similarity and the FNN library in R was used to build kNN models (Beygelzimer et al., 2015).

### 3.2   ML Modelling steps

A general equation was developed relating runoff to climatic variables as follow (equation 8):

$$R_t = f(P_t, P_{t-1}, P_{t-2}, \ldots, P_{t-lag}, Ta_t, Ta_{t-1}, Ta_{t-2}, \ldots, Ta_{t-lag}, W_t, W_{t-1}, W_{t-2}, \ldots, W_{t-lag}, Rh_t, Rh_{t-1}, Rh_{t-2}, \ldots, Rh_{t-lag})$$

$$\tag{8}$$

$R$ is green roof runoff, $P$ is precipitation, $Ta$ is air temperature, $W$ is wind speed, and $Rh$ is relative humidity. This is a simplification as the physical properties of the green roof also affect its runoff. However, using data from the same green roofs

in this study, Johannessen et al. (2018) found only a small variation in the hydrological performances between the different roof configurations and found the climatic variables to have high impacts on their performance. In the ML models in this study, climatic variables were lagged to represent the initial saturation of the green roofs at each time $t$. The values of $lag$ were optimized for each green roof and for each ML model during the process of hyperparameters optimization.

Data were aggregated into one-hour resolution, and snow accumulation periods were excluded (1 Oct. – 31 Mar.). The

data of each green roof were divided into three sets: training, validation and testing. The training datasets were used to train the parameters of the ML models. Validation datasets were used for hyperparameters optimization while the testing datasets

**Table 2.** Periods selected for model training, validation and testing

| City | Data | Period | | Amount of precipitation (mm) |
|------|------|--------|--------|------|
| | | From [dd.mm.yyyy] | To [dd.mm.yyyy] | |
| | Training | 01.04.2017 | 30.09.2017 | 1299.69 |
| Bergen | Validation | 01.04.2016 | 30.09.2016 | 1376.99 |
| | Testing | 01.04.2015 | 30.09.2015 | 936.32 |
| | Training | 01.04.2016 | 30.09.2016 | 429.82 |
| Oslo | Validation | 01.04.2017 | 30.09.2017 | 667.49 |
| | Testing | 01.04.2015 | 30.09.2015 | 551.61 |
| | Training | 01.04.2016 | 30.09.2016 | 727.50 |
| Sandnes | Validation | 01.04.2015 | 04.06.2015 | 299.98 |
| | Testing | 01.04.2017 | 30.09.2017 | 880.92 |
| | Training | 01.04.2017 | 30.09.2017 | 493.77 |
| Trondheim | Validation | 01.04.2016 | 30.09.2016 | 379.99 |
| | Testing | 01.04.2015 | 30.09.2015 | 546.62 |

were used for the independent evaluations of the ML models and for the comparisons with the conceptual models. The periods between 01.04.2015 to 30.09.2015 were used as testing datasets. At Sandnes, only data of two months in 2015 are available due to issues in the measurements. Hence, the periods of 01.04.2017 to 30.09.2017 were used as testing periods at Sandnes. Initially, the selection of the training periods was based on the amount of precipitation presented in table 2; the wettest year between 2016 to 2017 were initially selected as training periods. The rationale for the selection was that the wettest year covers a broader span of precipitation events which improves the generalization performance of the models. After the hyperparameters optimization, we further analyzed the change of ML performance when using the validations datasets for model training. Some of the validation datasets slightly improved the ML performance and hence were selected as training datasets. The final selection of the training, validation and testing periods is presented in table 2 .

### 3.2.1 ML hyperparameter tuning

ML models were tuned to achieve good modelling performance and to avoid overfitting. Hyperparameter tuning is the process of finding the optimal ML hyperparameters for the problem (e.g. number of hidden layers in ANN, number of LSTM units, k value in kNN, etc.). Bayesian optimization (BO) was selected for hyperparameters tuning (Snoek et al., 2012). This algorithm is suitable for functions in which evaluating one set of parameters is expensive and time-consuming. It was applied by Worland et al. (2018) to optimize hyperparameters for several machine learning models to predict low flows for ungauged basins. In

Bayesian optimization, the objective function (i.e. the relation between ML hyperparameters and the performance of the ML model in the validation data set) is approximated by a probabilistic model (e.g. Gaussian process) that is used to select the most promising hyperparameter to evaluate in the true objective function. The algorithm works as follow:

1. Select initial points of hyperparameters randomly and evaluate them in the true objective function.

2. Build a probabilistic model of the objective function (surrogate function) based on the initial points. Gaussian process was selected as the surrogate function of the objective function (Snoek et al., 2012; Worland et al., 2018).

3. Choose which hyperparameter to evaluate next in the true objective function based on the surrogate function by optimizing an acquisition function. The expected improvement (EI) was used as an acquisition function in this study (Snoek et al., 2012; Worland et al., 2018).

4. Use the new evaluated point to update the surrogate function.

5. Repeat steps 2-4 for N iterations

Prior to optimization, ML hyperparameters that require tuning and their upper and lower limits were selected (Table 3), following similar studies (Kratzert et al., 2019; Shortridge et al., 2016). For ANN and LSTM, dropout layers were implemented as a measure to reduce overfitting (Srivastava et al., 2014). At the dropout layer, a specific portion of the optimized weights and biases are set to zero randomly at each training epoch. This technique is used to prevent the network to learn specific pattern of the input noises and to focus on learning the general patterns of the data. For LSTM, only one hidden layer was selected for this study following the recommendation of the study of Ayzel (2019) in which, a grid search was performed for LSTM hyperparameters which compared thousands of LSTM structures. One hidden layer was found to perform reasonably well with lower computational cost compared with multiple hidden layers.

In the first step of the BO, random samples of hyperparemetrs (five in this study) were drawn from the selected ranges presented in table 3. These initial points were used to build a Gaussian process model. The Gaussian process model represents the objective function by constructing posteriors distribution of functions with high uncertainty bound far from the sampled points and low uncertainty bounds near the sampled points. In the next step, a continuous function (EI) is calculated for each point x along the Gaussian process model by determining two components. First, how much improvement is expected at x by comparing the mean of the Gaussian process model at the point x with the current best estimate from the sampled points. Second, how much is the uncertainty of the Gaussian process model at the point x, based on the uncertainly bounds. The point x that maximizes the value of EI is selected to be evaluated in the true objective function and the result is used to update the Gaussian process model for the next iteration. At the first iterations, the values of EI function are higher for regions with high uncertainty, so the algorithm fever points in new regions (exploration). After many iterations and new samples, the uncertainty bounds of the Gaussian process model decreases and the algorithm fevers areas with better solutions (exploitation). After N iterations (100 in this study), the algorithm returns the hyperparemeters that generate the best solution. In this study, the R library "ParBayesianOptimization" (Wilson, 2021) was used for the BO.

max

**Table 3.** Selected ML hyperparameters for tuning

| Models | Hyperparameters | Lower Limit | Upper Limit |
|--------|-----------------|-------------|-------------|
| ANN | Number of hidden layers | 1 | 4 |
| | Number of Neurons | 1 | 100 |
| | Dropout rate | 0 | 0.499 |
| | lag | 1 | 72 |
| LSTM | Number of hidden layers | 1 | 1 |
| | Number of LSTM units | 1 | 100 |
| | Dropout rate | 0 | 0.499 |
| | lag | 1 | 72 |
| M5 | Neighbors | 20 | 20 |
| | Committees | 20 | 20 |
| | lag | 1 | 72 |
| kNN | k | 1 | 100 |
| | lag | 1 | 72 |

### 255   3.3   The conceptual retention model

The sixteen roofs were modelled using a conceptual retention model (RM), which was developed and validated by Stovin et al. (2013). The RM model is intended to provide a robust tool that estimates green roof retention using simple water balance equations (equations 9,10 and 11).

$$R_t = \max(0, P_t - (S_{\max} - S_t) - \text{AET}_t) \tag{9}$$

$$S_t = min(S_{t-1} + P_t - \text{AET}_t, 0)) \tag{10}$$

$$\text{AET}_t = \text{PET}_t \times \frac{S_{t-1}}{S_{\max}} \tag{11}$$

$R_t$ is the runoff from a green roof at time $t$, $P_t$ is the precipitation at time $t$, $S_{\max}$ is the maximum storage available in a green roof and $S_t$ is the water stored in a green roof at time $t$. In our study region, Johannessen et al. (2018) found the Oudin's model for ET to be the most accurate for their water balance model and Almorox et al. (2015) recommended the use of Oudin for

cold climates. Hence, the potential evapotranspiration was computed using Oudin's model as follows (equations 12):

$$\text{PET}(\frac{\text{mm}}{\text{day}}) = \begin{cases} 0, & \text{if } Ta_{\text{mean}} \leq 5 \\ \frac{Ra}{\lambda\rho} \times 0.01 \times (Ta_{\text{mean}} + 5), & \text{if } Ta_{\text{mean}} > 5 \end{cases} \tag{12}$$

$Ta_{\text{mean}}$ is the daily mean temperature, $Ra$ is extra-terrestrial radiation derived from Julian day and latitude $(MJ.m^{-2})$, $\frac{1}{\lambda\rho} \approx 0.408$, $\lambda$ is the latent heat of water $(MJ.kg^{-1})$, $\rho$ is the volumetric mass of water $(kg.m^{-3})$.

The parameter $S_{\text{max}}$ represents the maximum retention capacity of the green roof or the difference between the field capacity and the permanent wilting point of the green roof substrate (Stovin et al., 2013). There exist standard laboratory tests to physically measure the substrate field capacity (Breuning and Yanders, 2008) and the permanent wilting point (Fassman and Simcock, 2012). In this study, however, $S_{\text{max}}$ values were estimated by assuming the field capacities of the roof layers from reported literature values as follow: vegetation mats were assumed to have 20% of the total substrate depth as a field capacity (Johannessen et al., 2018) , brick-based substrates were assumed to have 25 % of the total substrate depth as a field capacity (Stovin et al., 2013) while the drainage mats were assumed to have no permanent storage. The retention models with estimated $S_{\text{max}}$ is refereed to as uncalibrated retention models (RM$_{\text{uncalib}}$).

To allow for fair comparison with the ML models, retention models with calibrated $S_{\text{max}}$ values were used (RM$_{\text{calib}}$). For each roof, we ran the conceptual model by varying the value of $S_{\text{max}}$ between 10% to 50% of the substrate total depth. Values of $S_{\text{max}}$ that minimize the Volumetric error of the RM model were selected. The training periods in table 2 were selected for calibration.

### 3.4 ML Model evaluation

Methods were evaluated based on the performance on the testing datasets. With respect to retention estimation, flow accumulation curves were plotted for the simulated runoff from ML models against the observed runoff and compared with the results from the conceptual retention model. In addition, the percentage bias (PBIAS) values (equation 13) were calculated for each simulation for comparison. To evaluate the performance of ML models in estimating the temporal variation in runoff, the simulated runoff from ML models were plotted against the observed values and the NSE (equation 14) values were determined. Values of NSE > 0.5 were considered satisfactory (Moriasi et al., 2007; Rosa et al., 2015). To evaluate the potential of using ML as a useful tool for planning and design purposes, ML models were transferred between the roofs unchanged. The transferred models simulated the testing periods of each roof, and NSE was used to evaluate the transferability performance. Moreover, a volumetric factor (vol) based on the PBIAS was determined by using equation 15 to assess transferability in terms of volume estimation. A vol value of 1 indicates a perfect runoff volume estimation and hence a perfect retention estimation, while A vol value of zero indicates 100% error in volume estimation. Additionally, we compared the performance of transferred ML models with the uncalibrated retention models.

$$\text{PBIAS} = 100 \times \frac{\sum Q_{\text{obs}} - \sum Q_{\text{sim}}}{\sum Q_{\text{obs}}} \tag{13}$$

$$\mathrm{NSE} = 1 - \frac{\sum(Q_{\mathrm{obs}} - Q_{\mathrm{sim}})^2}{\sum(Q_{\mathrm{obs}} - \overline{Q_{\mathrm{obs}}})^2} \tag{14}$$

$$\mathrm{vol} = 1 - \frac{|\mathrm{PBIAS}|}{100} \tag{15}$$

## 4  Results and Discussion

### 4.1  Hyperparameter optimization (model tuning)

For LSTM and ANN models, the number of epochs was selected prior to the BO process by running an initial ANN model with 1000 epochs. After 30 epochs, the performance in the validation data didn't improve further and started to decrease after 100 epochs while improving in the training data set, which indicates overfitting. Therefore, 30 epochs were selected as an optimal value. Then, the BO algorithm was applied with 100 iterations for each ML model and for each roof. The selected hyperparameters of each iteration were stored. Figure 3 presents the empirical probability density distributions of the selected ANN hyperparemeters by the BO and their associated performances in the validation datasets. The results can interpreted as that hyperparmeters with high density values are located in regions that maximized the modelling performance in the validation datasets. The hyperparemeters that generated the best results at the validation datasets were selected for each roof and each ML model, as presented in table 4.

Based on figure 3, ANN with one hidden layer was found to be sufficient for most of the roofs in the study. Hence, deep ANN architectures , i.e. ANN models with many hidden layers, might not be required for this task. This has an important implication as deep ANN models are computationally expensive and prone to overfitting. Likewise, Ayzel (2019), found that deep LSTM models are not required for predicting runoff at hourly time steps, while Zhang et al. (2018) found a single-layer LSTM to perform better than LSTM model with two layers for predicting daily water level depths in agricultural land.

Another interesting finding is the lag values which are varied between the cities. It can be noted that, the lag values were smaller in Beren and Sandnes compared to Trondheim and Oslo. To interpret this finding, rainfall events, with 6-hour intra-event periods, were extracted from the three datasets at the four cities and compared, as shown in Figure 4. Bergen roofs received events with higher amount and duration compared to Oslo and Trondheim roofs, whereas the antecedent dry weather periods (ADWP) at Oslo and Trondheim are longer than BERG. Hence, due to the longer ADWP, a longer memory of the system is required to account for the wider range of possible initial saturation, compared to Bergen roofs.

### 4.2  Model evaluations

#### 4.2.1  Retention estimation

Machine learning models were built for all roofs based on the optimized hyperparemeters, selected by the BO algorithm. Figure 5 illustrates the simulated and observed runoff cumulative curves together with the cumulative precipitation for each

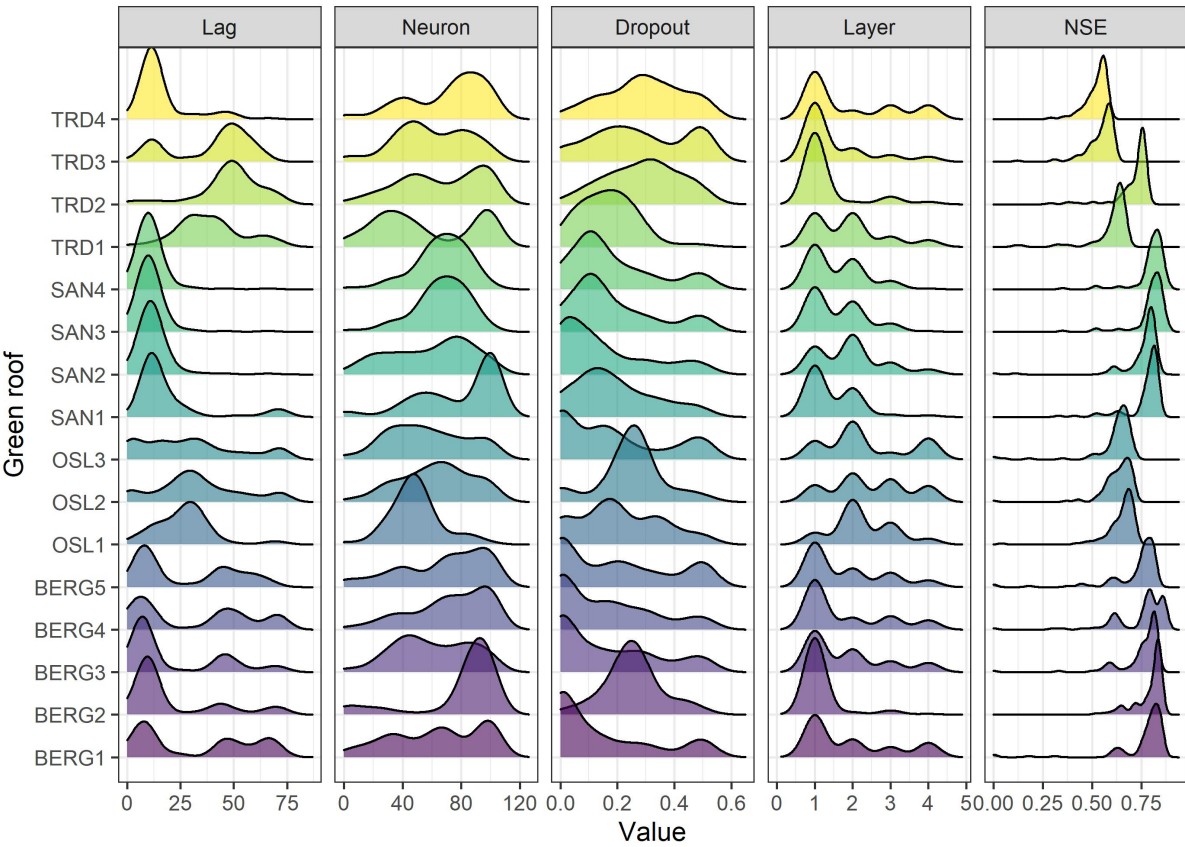

**Figure 3.** Empirical density distributions of the selected ANN hyperparemeters by the Bayesian optimization algorithm and their associated performances in the validation datasets

roof, and table 5 shows the values of PBIAS and NSE of the models at the testing datasets. The results presented in figure 5
and table 5 confirm that the ML models and the conceptual models can reproduce the observed runoff volume in most of the
green roofs. By comparing the median values of the PBIAS on the testing periods, LSTM yielded only -0.15% with a standard
deviation of 8.61%. Following LSTM, median values of -0.55%, -1.50% and 4.05% were obtained by the $RM_{calib}$, ANN and
$RM_{uncalib}$ models, respectively. The M5 models yielded simulation with a median PBIAS of -9.4% while the kNN yielded
the highest volumetric errors with a median PBIAS of -24.25% with a standard deviation of 9.78%. It can be noted that the
conceptual retention models and ML models, except kNN, could produce results that are classified as acceptable modelling
results regarding volumetric error (|PBIAS|<25%), as per Moriasi et al. (2007).

**Table 4.** Results of ML Hyperparameters tuning

| GR | ANN | | | | LSTM | | | M5 | | | kNN | |
|---|---|---|---|---|---|---|---|---|---|---|---|---|
| | Neurons | Layers | Dropout rate | Lag | Units | Dropout rate | Lag | Neighbors | committees | Lag | k | Lag |
| BERG1 | 100 | 1 | 0.00 | 5 | 100 | 0.19 | 5 | 6 | 18 | 14 | 6 | 4 |
| BERG2 | 89 | 1 | 0.25 | 9 | 88 | 0.36 | 7 | 9 | 40 | 13 | 8 | 2 |
| BERG3 | 43 | 2 | 0.00 | 6 | 85 | 0.50 | 45 | 6 | 12 | 19 | 8 | 3 |
| BERG4 | 100 | 1 | 0.00 | 5 | 72 | 0.00 | 4 | 5 | 88 | 48 | 8 | 3 |
| BERG5 | 70 | 2 | 0.07 | 7 | 73 | 0.34 | 47 | 6 | 100 | 43 | 7 | 6 |
| OSL1 | 53 | 2 | 0.17 | 31 | 73 | 0.38 | 18 | 3 | 60 | 44 | 11 | 5 |
| OSL2 | 41 | 2 | 0.26 | 34 | 100 | 0.22 | 13 | 9 | 18 | 48 | 10 | 4 |
| OSL3 | 38 | 2 | 0.12 | 38 | 83 | 0.50 | 18 | 3 | 12 | 43 | 11 | 4 |
| SAN1 | 100 | 1 | 0.16 | 10 | 100 | 0.00 | 36 | 9 | 54 | 36 | 11 | 7 |
| SAN2 | 39 | 2 | 0.12 | 8 | 70 | 0.29 | 2 | 5 | 45 | 48 | 8 | 4 |
| SAN3 | 70 | 2 | 0.10 | 9 | 100 | 0.21 | 10 | 1 | 100 | 29 | 10 | 7 |
| SAN4 | 70 | 2 | 0.10 | 9 | 100 | 0.21 | 10 | 1 | 100 | 27 | 35 | 4 |
| TRD1 | 97 | 2 | 0.27 | 61 | 55 | 0.07 | 48 | 1 | 26 | 27 | 7 | 16 |
| TRD2 | 99 | 1 | 0.33 | 50 | 35 | 0.50 | 48 | 6 | 100 | 45 | 5 | 16 |
| TRD3 | 91 | 1 | 0.29 | 53 | 31 | 0.00 | 46 | 9 | 100 | 46 | 28 | 5 |
| TRD4 | 70 | 4 | 0.27 | 13 | 100 | 0.24 | 48 | 6 | 20 | 32 | 47 | 4 |

### 4.2.2 Temporal variations in runoff

Table 5 presents the NSE values for training and validation periods for the ML models. Most ML models yielded satisfactory results in the testing periods (NSE > 0.5). M5 models produced results with highest NSE values, with a median value of 0.80. Both ANN and LSTM produced result with a median NSE values of 0.67. Figure 6 shows the observed and simulated hydrographs for BERG2 roof, which confirms the ability of the ML models to reproduce the observed runoff. In contrast, the conceptual models produced satisfactory results in only five roofs. We found the green roofs in our study to detain small and medium sized events for up to two hours. The conceptual model failed to simulate these dynamics due to lack of routing.

The performance of ML models varied between the different cities more than between the different configurations. Johannessen et al. (2018), using the same data in this study, observed similar hydrological performance for the different configuration within the same city. It should be noted that, however, the geometries of the roofs are identical at each city (Table 1). The performance of the ML methods can be explained based on this comparison between the cities' rainfall characteristics (Figure 4). For instance, the NSE values of the ML models are higher in Bergen roofs in comparisons to Oslo roofs. As mentioned earlier, Oslo roofs have a wider range of possible initial saturations. Therefore, one year of training data might not be enough to cover

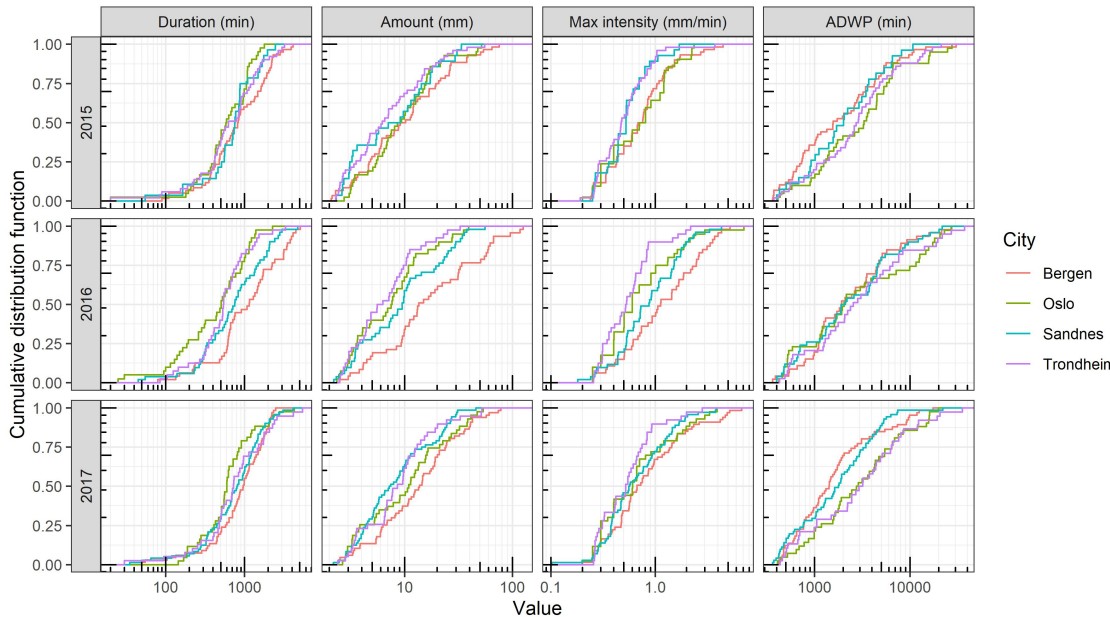

**Figure 4.** Comparison between the rainfall events at the four Norwegian cities

this wide range of runoff possibilities. On the other hand, Bergen roofs received more frequent and intense precipitation events resulting in a small range of possibilities of initial saturation that could be covered using one year only. The kNN method produced lower NSE values compared to the other ML models. This was attributed to the relatively small training data used in this study as kNN estimates the performance depending on the similarity to the previous time steps.

LSTM maintains a state value between consecutive time steps which makes it more suitable for modelling green roofs where initial saturation plays an important role in green roof runoff generation process. A comparison was made between ANN and LSTM at TRD1 (Figure 7) to demonstrate the potential of LSTM. ANN was found to produce runoff when no precipitation occurred, unlike LSTM. Moreover, LSTM could simulate the flow peaks more accurately than ANN. Likewise, Kratzert et al. (2018) found LSTM simulations to be smoother than a normal recurrent neural network and to be better in accounting for the storage capacity (including snow accumulation) of a natural catchment

### 4.2.3 Effect of training data and ensemble modelling

The performance of ML models when using different data for model training was evaluated. For each roof, two ML models were built; one by using the training datasets in table 2 for model training and one by using the validation datasets in table 2 for model training. Sandnes roofs were excluded from this analysis due the missing data in one 2015, as discussed earlier. Figure 8 demonstrates the performance of LSTM models at BERG2, OSL1 and TRD1 roofs when using different data for model training. The performances of the two LSTM models (LSTM1 and LSTM2) were quite similar, as presented in figure 8. One idea that could improve the estimates of the ML models is to combine the simulations from several ML models that are build

**Table 5.** Overall modelling performance (testing data)

| GR | ANN | | LSTM | | kNN | | M5 | | RM$_{calib}$ | | RM$_{uncalib}$ | |
|---|---|---|---|---|---|---|---|---|---|---|---|---|
| | NSE | PBIAS | NSE | PBIAS | NSE | PBIAS | NSE | PBIAS | NSE | PBIAS | NSE | PBIAS |
| BERG1 | 0.72 | 9.80 | 0.75 | 9.00 | 0.76 | -5.40 | 0.84 | 7.40 | 0.20 | 8.10 | 0.14 | 12.5 |
| BERG2 | 0.83 | -0.20 | 0.82 | -4.20 | 0.77 | -6.70 | 0.91 | -8.80 | 0.35 | -11.00 | 0.33 | -9 |
| BERG3 | 0.76 | -9.70 | 0.78 | -8.30 | 0.74 | -11.00 | 0.84 | -10.50 | 0.61 | -9.00 | 0.61 | -9.7 |
| BERG4 | 0.81 | -7.60 | 0.81 | -10.90 | 0.82 | -18.40 | 0.89 | -13.30 | 0.61 | -8.90 | 0.63 | -15.4 |
| BERG5 | 0.66 | 3.30 | 0.64 | 1.80 | 0.66 | -10.80 | 0.72 | -5.90 | -0.01 | 0.80 | -0.19 | 17.5 |
| OSL1 | 0.58 | -5.40 | 0.61 | -3.90 | 0.51 | -24.70 | 0.61 | -10.00 | 0.53 | 6.60 | 0.53 | 6.6 |
| OSL2 | 0.54 | -17.60 | 0.60 | -3.70 | 0.49 | -26.90 | 0.55 | -14.00 | 0.76 | -7.20 | 0.8 | -13 |
| OSL3 | 0.58 | 5.60 | 0.60 | 2.70 | 0.53 | -21.60 | 0.65 | -3.60 | 0.44 | 10.50 | 0.51 | 3.5 |
| SAN1 | 0.83 | 9.50 | 0.70 | -3.10 | 0.75 | -32.60 | 0.90 | -6.50 | -0.10 | -1.50 | -0.3 | 4.6 |
| SAN2 | 0.73 | -3.00 | 0.67 | -3.00 | 0.67 | -27.70 | 0.80 | -15.00 | 0.39 | -9.40 | 0.41 | -10.6 |
| SAN3 | 0.47 | -1.50 | 0.46 | 1.60 | 0.49 | -29.40 | 0.65 | -6.40 | -0.34 | 0.80 | -0.5 | 7.1 |
| SAN4 | 0.47 | -1.50 | 0.46 | 1.60 | 0.46 | -28.10 | 0.64 | -6.50 | -0.34 | 0.80 | -0.5 | 7.1 |
| TRD1 | 0.73 | -12.70 | 0.75 | 5.80 | 0.58 | -36.40 | 0.81 | -13.70 | -0.51 | 10.50 | -1.71 | 38 |
| TRD2 | 0.75 | -0.90 | 0.74 | -1.90 | 0.59 | -30.80 | 0.79 | -11.80 | -0.32 | -5.40 | -0.76 | 6.6 |
| TRD3 | 0.69 | 5.20 | 0.66 | 10.70 | 0.55 | -11.40 | 0.80 | -7.80 | -0.10 | 0.40 | -0.08 | -1.4 |
| TRD4 | 0.64 | -16.40 | 0.69 | 25.20 | 0.52 | -23.80 | 0.83 | -11.00 | 0.38 | -5.10 | 0.38 | 2.5 |
| Median | 0.71 | -1.50 | 0.68 | -0.15 | 0.58 | -24.25 | 0.80 | -9.40 | 0.28 | -0.55 | 0.24 | 4.05 |
| Mean | 0.67 | -2.69 | 0.67 | 1.21 | 0.62 | -21.61 | 0.76 | -8.59 | 0.16 | -1.03 | 0.28 | 3.78 |
| sd | 0.12 | 8.47 | 0.11 | 8.61 | 0.12 | 9.78 | 0.11 | 5.43 | 0.41 | 7.25 | 0.66 | 13.40 |

from different datasets (ensemble modelling). As shown in figure 8, the average simulations of two LSTM models yielded better simulation for the testing dataset (lower PBIAS and slightly higher NSE values) in comparison to the estimate of each of the two LSTM models . However, this is only true when the two LSTM models have positive and negative biases. Nevertheless, we found few green roofs where the two LSTM models from the two years resulted in biases of the same direction (at BERG3 and OSL3 roofs).

### 4.2.4 Transferability

Models were transferred between the roofs unchanged to simulate the testing periods. Figure 9 presents the transferability performance measured by NSE. Some models could yield satisfactory results in different locations (NSE > 0.5). For instance, M5 models that were trained using data from Bergen roofs could yield satisfactory performance in almost all cities. Figure 10 presents an example of transferred M5 models between BERG2, OSL3,SAN1 and TRD1 roofs. It can be noted that models that

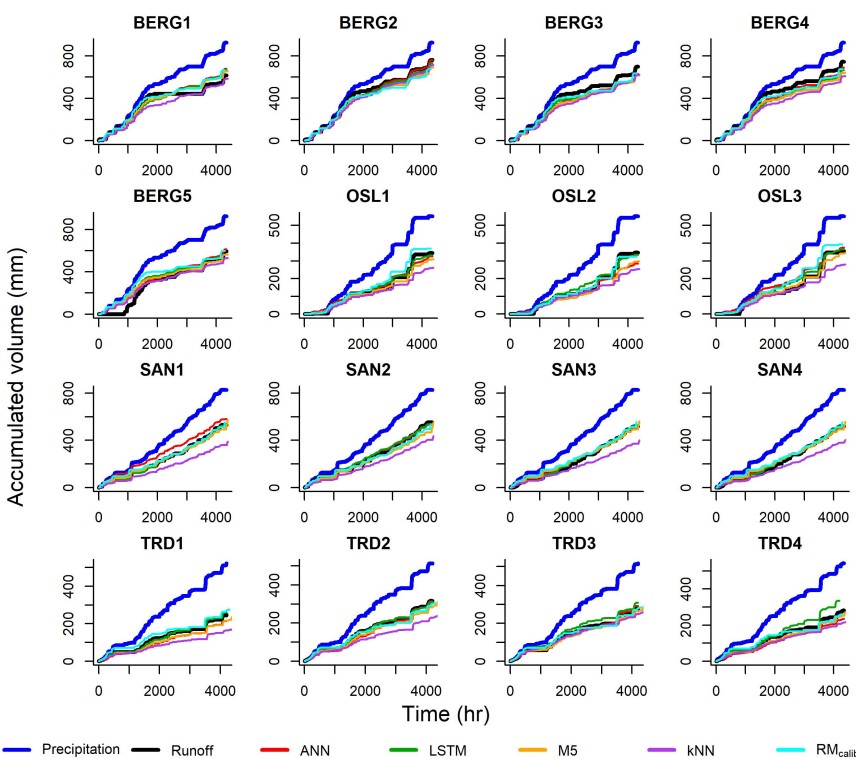

**Figure 5.** Cumulative precipitation, observed and simulated runoff of the green roofs

are trained in wetter cities, such as Bergen, overestimated the flows at cities with lower precipitation, such as Trondheim. Figure 11 presents the transferability performance with respect to retention estimation measured by vol. Transferred ML model could simulate result with acceptable accuracy (vol>0.75) (Moriasi et al., 2007) between Trondheim and Oslo cities and between

375 Sandnes and Bergen cities with some exceptions. This can be somewhat attributed to the similarity in climatic conditions between the cities (Figure 4). However, the uncalibrated conceptual models in this study could produce better volume estimates than the transferred ML models in most cases. This implies that using the conceptual model with literature estimates of the $S_{\mathrm{max}}$ parameter is preferable over the transferred ML to estimate the annual retention for new roofs.

## 4.3 Machine learning potentials for green roof hydrological modelling

The present paper has demonstrated that well-trained ML models can be applied to estimate retention process (rainfall losses) in a range of different green roof systems. The predictions are comparable in accuracy to a conceptual water balance model based on losses due to evapotranspiration. Additionally, well-trained ML models showed more accurate predictions of runoff hydrographs than the conceptual water balance model which is encouraging for detention modelling. Moreover, aggregating the simulations of many ML models (ensamble modelling) appears to improve the prediction and can be investigated in fu-

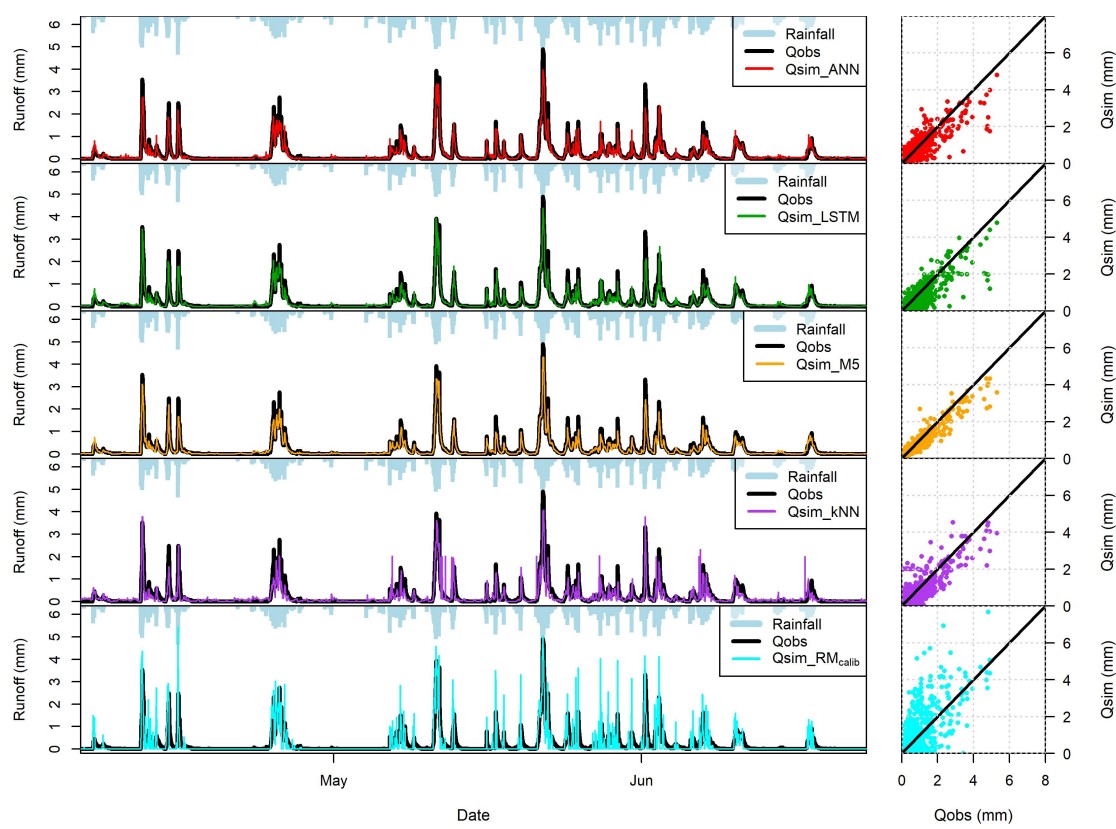

**Figure 6.** Performance of ML models on the testing period (BERG 2). The hydrographs were plotted for around three months period (2000 hours), while the Q-Q plots were plotted for the entire testing period

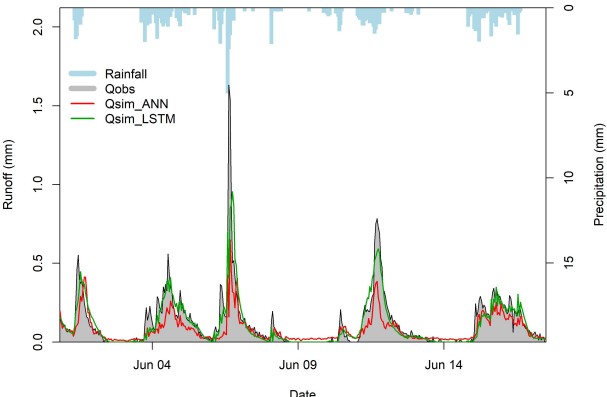

**Figure 7.** Comparison between LSTM and ANN at TRD1 roof

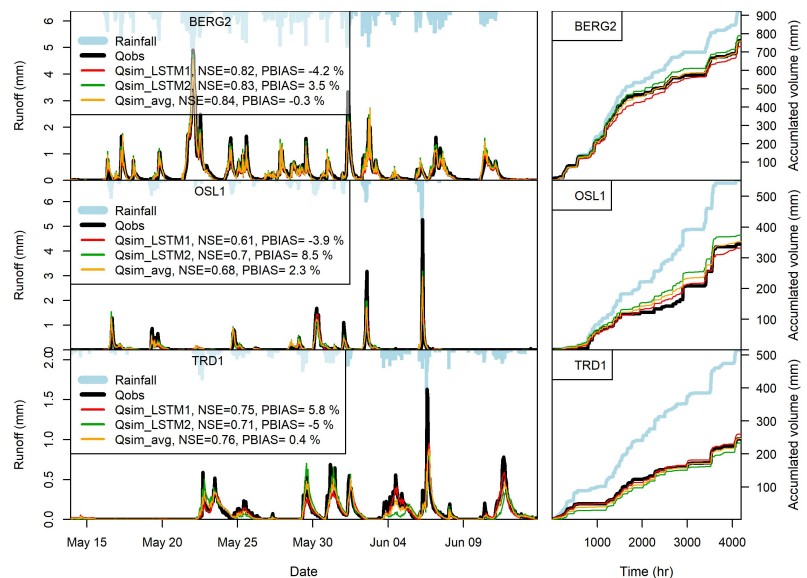

**Figure 8.** The performance of LSTM models in the testing datasets at BERG2,OSL1 AND TRD1 roofs when using different data for model training. LSTM1 is trained using the testing period presented in table 2, While LSTM2 is trained using the validation period presented in table 2. Qsim-avg is the average of Qsim-LSTM1 and Qsim-LSTM2

ture studies. Detention modelling is required to estimate the lag and attenuation of runoff associated with any rainfall that is not retained. In practice, many modelling frameworks rely on calibrated reservoir routing models to estimate the cumulative detention effects of multiple interacting component layers, and few (if any) convincing validation cases for a complete detention modelling framework have been presented. It would therefore be very valuable to explore whether the ML models, when trained on higher temporal resolution datasets, have the capability to capture these complex detention effects better than the

alternative black-box approaches.

## 5    Conclusions

Four machine learning models, commonly used in runoff modelling studies, were applied to simulate runoff from sixteen green roofs located in four Norwegian cities with different climatic conditions. We further investigated the potential of using ML models to estimate performance of new roofs where runoff data are not available for model training. This was done by

395 means of transferring ML models between the roofs in the study. Our results confirms the ability of well-trained ML models to estimate green roof retention and the temporal runoff dynamics.The estimates of the annual retention were comparable to a proven conceptual model. Despite the 1-hr time step, the ML models provided accurate simulations of runoff dynamics i.e discharge hydrographs (NSE values higher than 0.5 in most cases) which is encouraging for detention modelling. The LSTM demonstrated better modelling performance by maintaining a state value between consecutive time step, which makes it more

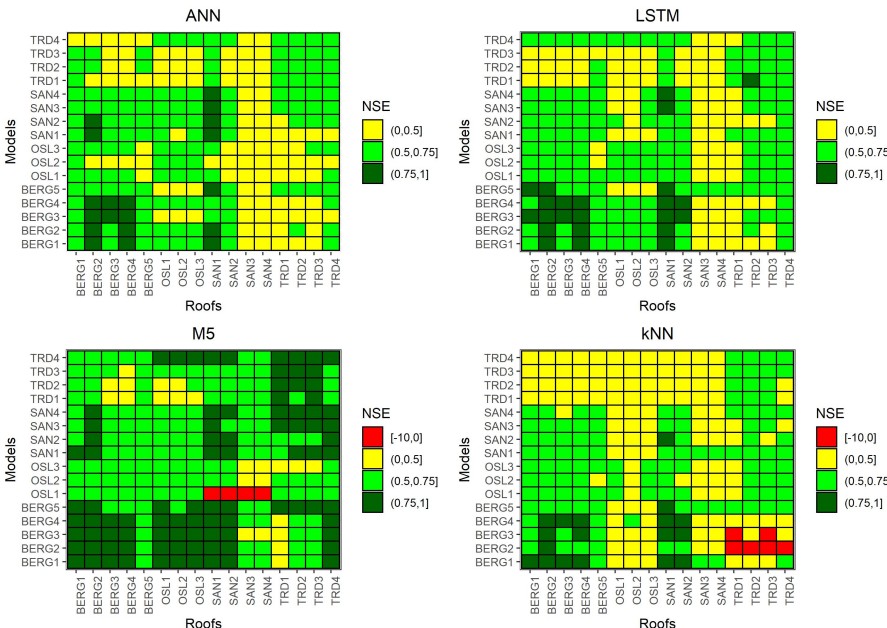

**Figure 9.** Transferability between the different roofs (NSE). Models in the y-axis are used to simulate the measured green roofs in the x-axis

appropriate for simulating runoff of green roofs. In future studies, shorter time-steps will be applied to estimate detention metrics.

Some transferred ML models could give acceptable model performance (NSE > 0.5, |PBIAS|<25%) in different locations. However, we recommend using the conceptual model with literature values of the $S_{\max}$ parameter to estimate the annual retention of new roofs over the transferred model as it give accurate volume estimations.

*Author contributions.* Elhadi was responsible of the machine learning models, including the selection of the four models, hyperparameter tuning and and models training. Virginia Stovin proposed the comparison with the conceptual retention model. Vincent Pons assisted Elhadi during ML modelling and the conceptual model application. Knut and Tone supervised each step of the research from the start. Elhadi wrote the first manuscript which was revised many times by all co-authors

*Competing interests.* The authors declare that they have no competing interests

*Acknowledgements.* The authors would like to acknowledge the financial support by the Research Council of Norway through the Centre for Research-based Innovation "Klima 2050" (www.klima2050.no).

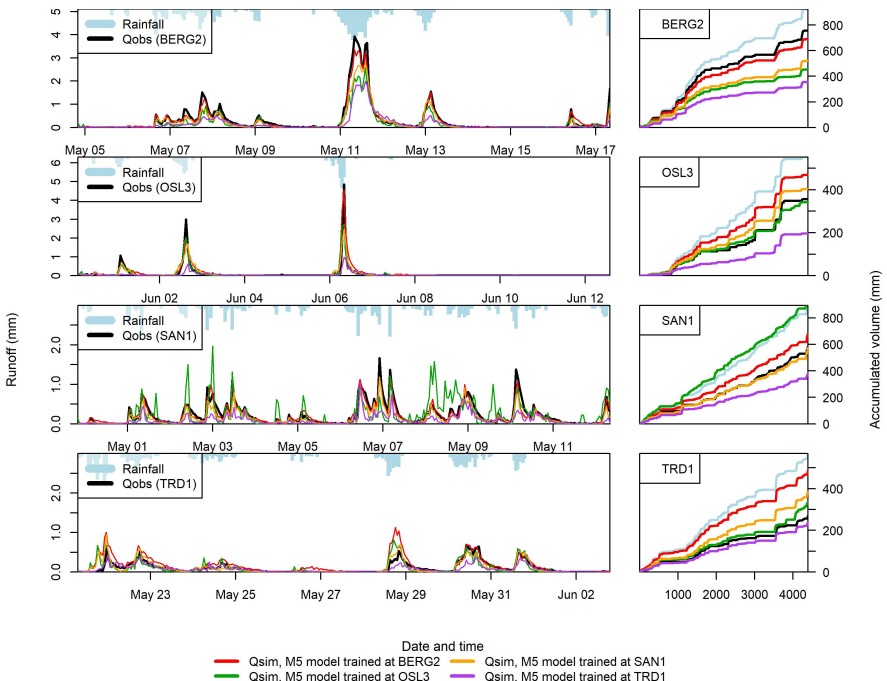

**Figure 10.** The performance of the transferred ML models at BERG2, OSL3,SAN1 and TRD1. The hydrographs were plotted for selected periods, while the cumulative plots were plotted for the entire testing period

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

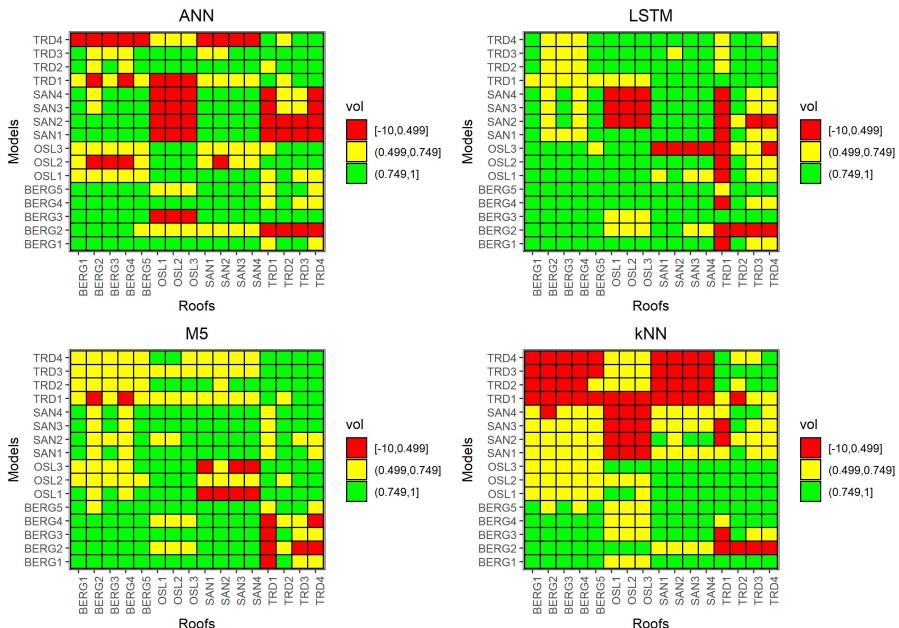

**Figure 11.** Transferability between the different roofs (vol). Models in the y-axis are used to simulate the measured green roofs in the x-axis

Bouzouidja, R., Séré, G., Claverie, R., Ouvrard, S., Nuttens, L., and Lacroix, D.: Green roof aging: Quantifying the impact of substrate evolution on hydraulic performances at the lab-scale, Journal of Hydrology, 564, 416–423, 2018.

Breuning, J. and Yanders, A.: FLL guidelines for the planning, construction and maintenance of green roofing, 2008.

Brunetti, G., Šimnek, J., and Piro, P.: A comprehensive analysis of the variably saturated hydraulic behavior of a green roof in a mediterranean climate, Vadose Zone Journal, 15, 1–17, 2016.

Carson, T., Marasco, D., Culligan, P., and McGillis, W.: Hydrological performance of extensive green roofs in New York City: observations and multi-year modeling of three full-scale systems, Environmental Research Letters, 8, 024 036, 2013.

Chollet, F. et al.: keras, 2015.

Cipolla, S. S., Maglionico, M., and Stojkov, I.: A long-term hydrological modelling of an extensive green roof by means of SWMM, Ecological Engineering, 95, 876–887, 2016.

Daniell, T.: Neural networks. Applications in hydrology and water resources engineering, in: National Conference Publication- Institute of Engineers. Australia, 1991.

DHI: MIKE URBAN Collection System. Modelling of Storm Water Drainage Networks and Sewer Collection Systems. User Guide, Danish Hydraulic Institute (DHI), Hørsholm, Denmark, 2017.

Dunnett, N. and Kingsbury, N.: Planting Green Roofs and Living Walls, 2004.

Fassman, E. and Simcock, R.: Moisture measurements as performance criteria for extensive living roof substrates, Journal of Environmental Engineering, 138, 841–851, 2012.

Fassman-Beck, E., Voyde, E., Simcock, R., and Hong, Y. S.: 4 Living roofs in 3 locations: Does configuration affect runoff mitigation?, Journal of Hydrology, 490, 11–20, 2013.

Gharaei-Manesh, S., Fathzadeh, A., and Taghizadeh-Mehrjardi, R.: Comparison of artificial neural network and decision tree models in estimating spatial distribution of snow depth in a semi-arid region of Iran, Cold Regions Science and Technology, 122, 26–35, 2016.

Goyal, M. K., Ojha, C., Singh, R., Swamee, P., Nema, R., et al.: Application of ANN, fuzzy logic and decision tree algorithms for the development of reservoir operating rules, Water resources management, 27, 911–925, 2013a.

Goyal, M. K., Ojha, C., Singh, R., Swamee, P., et al.: Application of artificial neural network, fuzzy logic and decision tree algorithms for modelling of streamflow at Kasol in India, Water science and technology, 68, 2521–2526, 2013b.

Hernes, R. R., Gragne, A. S., Abdalla, E. M., Braskerud, B. C., Alfredsen, K., and Muthanna, T. M.: Assessing the effects of four SUDS scenarios on combined sewer overflows in Oslo, Norway: evaluating the low-impact development module of the Mike Urban model, Hydrology Research, 51, 1437–1454, 2020.

Hochreiter, S. and Schmidhuber, J.: Long short-term memory, Neural computation, 9, 1735–1780, 1997.

Hsu, K.-l., Gupta, H. V., and Sorooshian, S.: Artificial neural network modeling of the rainfall-runoff process, Water resources research, 31, 2517–2530, 1995.

Hu, C., Wu, Q., Li, H., Jian, S., Li, N., and Lou, Z.: Deep learning with a long short-term memory networks approach for rainfall-runoff simulation, Water, 10, 1543, 2018.

Jahanfar, A., Drake, J., Sleep, B., and Gharabaghi, B.: A modified FAO evapotranspiration model for refined water budget analysis for Green Roof systems, Ecological engineering, 119, 45–53, 2018.

Javan, K., Lialestani, M. R. F. H., and Nejadhossein, M.: A comparison of ANN and HSPF models for runoff simulation in Gharehsoo River watershed, Iran, Modeling Earth Systems and Environment, 1, 1–13, 2015.

Johannessen, B. G., Hanslin, H. M., and Muthanna, T. M.: Green roof performance potential in cold and wet regions, Ecological Engineering, 106, 436–447, 2017.

Johannessen, B. G., Muthanna, T. M., and Braskerud, B. C.: Detention and retention behavior of four extensive green roofs in three nordic climate zones, Water, 10, 671, 2018.

Johannessen, B. G., Hamouz, V., Gragne, A. S., and Muthanna, T. M.: The transferability of SWMM model parameters between green roofs with similar build-up, Journal of Hydrology, 569, 816–828, 2019.

Karlsson, M. and Yakowitz, S.: Nearest-neighbor methods for nonparametric rainfall-runoff forecasting, Water Resources Research, 23, 1300–1308, 1987.

Kottek, M., Grieser, J., Beck, C., Rudolf, B., and Rubel, F.: World map of the Köppen-Geiger climate classification updated, 2006.

Kratzert, F., Klotz, D., Brenner, C., Schulz, K., and Herrnegger, M.: Rainfall–runoff modelling using long short-term memory (LSTM) networks, Hydrology and Earth System Sciences, 22, 6005–6022, 2018.

Kratzert, F., Klotz, D., Shalev, G., Klambauer, G., Hochreiter, S., and Nearing, G.: Towards learning universal, regional, and local hydrological behaviors via machine learning applied to large-sample datasets, Hydrology and Earth System Sciences, 23, 5089–5110, 2019.

Krebs, G., Kuoppamäki, K., Kokkonen, T., and Koivusalo, H.: Simulation of green roof test bed runoff, Hydrological processes, 30, 250–262, 2016.

Kuhn, M., Weston, S., Keefer, C., and Coulter, N.: Cubist models for regression, R package Vignette R package version 0.0, 18, 2012.

Li, S., Kazemi, H., and Rockaway, T. D.: Performance assessment of stormwater GI practices using artificial neural networks, Science of the total environment, 651, 2811–2819, 2019.

Li, Y. and Babcock Jr, R. W.: Modeling hydrologic performance of a green roof system with HYDRUS-2D, Journal of Environmental Engineering, 141, 04015 036, 2015.

Liu, R. and Fassman-Beck, E.: Hydrologic response of engineered media in living roofs and bioretention to large rainfalls: experiments and modeling, Hydrological Processes, 31, 556–572, 2017.

Liu, X. and Chui, T. F. M.: Evaluation of green roof performance in mitigating the impact of extreme storms, Water, 11, 815, 2019.

Modaresi, F., Araghinejad, S., and Ebrahimi, K.: A comparative assessment of artificial neural network, generalized regression neural network, least-square support vector regression, and K-nearest neighbor regression for monthly streamflow forecasting in linear and nonlinear

conditions, Water Resources Management, 32, 243–258, 2018.

Moriasi, D. N., Arnold, J. G., Van Liew, M. W., Bingner, R. L., Harmel, R. D., and Veith, T. L.: Model evaluation guidelines for systematic quantification of accuracy in watershed simulations, Transactions of the ASABE, 50, 885–900, 2007.

Multiphysics, C.: User Guide Version 4.4, COMSOL Multiphysics, Stockholm, Sweden, 2013.

Palla, A., Gnecco, I., and Lanza, L. G.: Unsaturated 2D modelling of subsurface water flow in the coarse-grained porous matrix of a green

roof, Journal of Hydrology, 379, 193–204, 2009.

Palla, A., Gnecco, I., and Lanza, L.: Compared performance of a conceptual and a mechanistic hydrologic models of a green roof, Hydrological Processes, 26, 73–84, 2012.

Peng, Z. and Stovin, V.: Independent validation of the SWMM green roof module, Journal of Hydrologic Engineering, 22, 04017 037, 2017.

Peng, Z., Smith, C., and Stovin, V.: Internal fluctuations in green roof substrate moisture content during storm events: Monitored data and

model simulations, Journal of Hydrology, 573, 872–884, 2019.

Quinlan, J. R.: Combining instance-based and model-based learning, in: Proceedings of the tenth international conference on machine learning, pp. 236–243, 1993.

Quinlan, J. R. et al.: Learning with continuous classes, in: 5th Australian joint conference on artificial intelligence, vol. 92, pp. 343–348, World Scientific, 1992.

Radfar, A. and Rockaway, T. D.: Captured runoff prediction model by permeable pavements using artificial neural networks, Journal of Infrastructure Systems, 22, 04016 007, 2016.

Rezaei, F., Jarrett, A., Berghage, R., and Beattie, D.: Evapotranspiration rates from extensive green roof plant species, in: 2005 ASAE Annual Meeting, p. 1, American Society of Agricultural and Biological Engineers, 2005.

Rosa, D. J., Clausen, J. C., and Dietz, M. E.: Calibration and verification of SWMM for low impact development, JAWRA Journal of the

American Water Resources Association, 51, 746–757, 2015.

Rossman, L. A. et al.: Storm water management model user's manual, version 5.0, National Risk Management Research Laboratory, Office of Research and . . . , 2010.

Rumelhart, D. E., Hinton, G. E., and Williams, R. J.: Learning representations by back-propagating errors, nature, 323, 533–536, 1986.

She, N. and Pang, J.: Physically based green roof model, Journal of hydrologic engineering, 15, 458–464, 2010.

Shen, C.: A transdisciplinary review of deep learning research and its relevance for water resources scientists, Water Resources Research, 54, 8558–8593, 2018.

Sherrard Jr, J. A. and Jacobs, J. M.: Vegetated roof water-balance model: experimental and model results, Journal of Hydrologic Engineering, 17, 858–868, 2012.

Shortridge, J. E., Guikema, S. D., and Zaitchik, B. F.: Machine learning methods for empirical streamflow simulation: a comparison of model

accuracy, interpretability, and uncertainty in seasonal watersheds, Hydrology and Earth System Sciences, 20, 2611–2628, 2016.

Sims, A. W., Robinson, C. E., Smart, C. C., and O'Carroll, D. M.: Mechanisms controlling green roof peak flow rate attenuation, Journal of Hydrology, 577, 123 972, 2019.

Simunek, J., Vogel, T., and van Genuchten, M. T.: The SWMS_2D code for simulating water flow and solute transport in two-dimensional variably saturated media, US Salinity Laboratory, Agricultural Research Service, US Department of . . . , 1994.

Simunek, J., Van Genuchten, M. T., and Sejna, M.: The HYDRUS-1D software package for simulating the one-dimensional movement of water, heat, and multiple solutes in variably-saturated media, University of California-Riverside Research Reports, 3, 1–240, 2005.

Snoek, J., Larochelle, H., and Adams, R. P.: Practical bayesian optimization of machine learning algorithms, Advances in neural information processing systems, 25, 2012.

Solomatine, D. P. and Dulal, K. N.: Model trees as an alternative to neural networks in rainfall—runoff modelling, Hydrological Sciences
Journal, 48, 399–411, 2003.

Soulis, K. X., Valiantzas, J. D., Ntoulas, N., Kargas, G., and Nektarios, P. A.: Simulation of green roof runoff under different substrate depths and vegetation covers by coupling a simple conceptual and a physically based hydrological model, Journal of environmental management, 200, 434–445, 2017.

Srivastava, N., Hinton, G., Krizhevsky, A., Sutskever, I., and Salakhutdinov, R.: Dropout: a simple way to prevent neural networks from
535 overfitting, The journal of machine learning research, 15, 1929–1958, 2014.

Stovin, V.: The potential of green roofs to manage urban stormwater, Water and Environment Journal, 24, 192–199, 2010.

Stovin, V., Poë, S., and Berretta, C.: A modelling study of long term green roof retention performance, Journal of environmental management, 131, 206–215, 2013.

Tokar, A. S. and Johnson, P. A.: Rainfall-runoff modeling using artificial neural networks, Journal of Hydrologic Engineering, 4, 232–239,
1999.

Tsang, S. and Jim, C. Y.: Applying artificial intelligence modeling to optimize green roof irrigation, Energy and Buildings, 127, 360–369, 2016.

Vesuviano, G. and Stovin, V.: A generic hydrological model for a green roof drainage layer, Water Science and Technology, 68, 769–775, 2013.

Vesuviano, G., Sonnenwald, F., and Stovin, V.: A two-stage storage routing model for green roof runoff detention, Water Science and Technology, 69, 1191–1197, 2014.

Wilson, S.: ParBayesianOptimization: Parallel Bayesian Optimization of Hyperparameters, https://CRAN.R-project.org/package= ParBayesianOptimization, r package version 1.2.4, 2021.

Worland, S. C., Farmer, W. H., and Kiang, J. E.: Improving predictions of hydrological low-flow indices in ungaged basins using machine
learning, Environmental modelling & software, 101, 169–182, 2018.

Wu, C., Chau, K. W., and Li, Y. S.: Predicting monthly streamflow using data-driven models coupled with data-preprocessing techniques, Water Resources Research, 45, 2009.

Yilmaz, A. G. and Muttil, N.: Runoff estimation by machine learning methods and application to the Euphrates Basin in Turkey, Journal of Hydrologic Engineering, 19, 1015–1025, 2014.

Yio, M. H., Stovin, V., Werdin, J., and Vesuviano, G.: Experimental analysis of green roof substrate detention characteristics, Water Science and Technology, 68, 1477–1486, 2013.

Young, C.-C., Liu, W.-C., and Wu, M.-C.: A physically based and machine learning hybrid approach for accurate rainfall-runoff modeling during extreme typhoon events, Applied Soft Computing, 53, 205–216, 2017.

Zhang, J., Zhu, Y., Zhang, X., Ye, M., and Yang, J.: Developing a Long Short-Term Memory (LSTM) based model for predicting water table
depth in agricultural areas, Journal of hydrology, 561, 918–929, 2018.