# Peer review of "Evaluating different machine learning methods to simulate runoff from extensive green roofs"

_Hydrology and Earth System Sciences, 2021_

## Author Comment (AC1)

Dear Referee,

We would like to thank you for the thoughtful comments which will contribute towards improving the manuscript.

This paper compares the performance of four machine learning algorithms (including a deep learning one) in simulating runoff from green roofs, and provides their benchmarking by also utilizing a conceptual model. The comparison is conducted by using data from sixteen green roofs located in four Norwegian cities, and the comparted algorithms are the Artificial Neural Network (ANN), M5 Model tree, Long Short-Term Memory (LSTM) and k-Nearest Neighbour (kNN) ones. Additional investigations focus on the transferability of the algorithms between different green roofs. The results show that the performance of the investigated algorithms is acceptable; however, the conceptual model should be preferred over the transferred machine and deep learning algorithms.

General comments

Overall, I believe that the paper is meaningful, interesting and mostly well-written with room for improvements.

We appreciate the positive comment about the study

Although my comments are quite few, I recommend major revisions, as the suggested improvements (mainly those prescribed with specific comment #1) are both important and necessary, to my view, for the model comparison (and the entire paper) to reach the best possible shape.

Specific comments

1)   In line 246, it is written that the "methods were evaluated based on the performance on the validation data sets". However, in line 221 it is written that "to avoid overfitting, the performance of changing hyperparameters was observed in the validation periods". As the validation set has been used for hyperparameter selection (i.e., for identifying the best version of its machine learning algorithm), the addition of an extra independent set (i.e., a test set that is not used for model selection) is necessary here. This extra set will serve the independent comparison between machine learning algorithms, as well as the independent comparison between machine learning algorithms and the conceptual model. Therefore, the datasets should be divided into (at least) three independent sets (including different data points), i.e., the training, validation and test sets.

We thank the reviewer for this thoughtful comment which will be included in the revised manuscript. We would like first to clarify the old setup that we applied in the study. We only optimized ML hyperparameters for 4 roofs (one roof in each city). The validation data sets of the four selected roofs were used for model selection (i.e. hyperparameter tuning). For the other 12 roofs in the study, the validation data sets were not used for model selection. However, we agree with point #3 that a hyperparameter tuning should be done for each roof. Therefore, we will modify the results by dividing

our data into three sets for model training, hyperparameter tuning and comparisons as suggested by the reviewer (Figure 1)

| old setup | | | | new setup | | | |
|---|---|---|---|---|---|---|---|
| **GR** | **Data set** | | | **GR** | **Data set** | | |
| | **Training** | **Validation** | **Testing** | | **Training** | **Validation** | **Testing** |
| BERG1 | 2017 | 2016 | 2015 | BERG1 | 2017 | 2016 | 2015 |
| BERG2 | 2017 | 2016 | 2015 | BERG2 | 2017 | 2016 | 2015 |
| BERG3 | 2017 | 2016 | 2015 | BERG3 | 2017 | 2016 | 2015 |
| BERG4 | 2017 | 2016 | 2015 | BERG4 | 2017 | 2016 | 2015 |
| BERG5 | 2017 | 2016 | 2015 | BERG5 | 2017 | 2016 | 2015 |
| OSL1 | 2015 | 2017 | 2016 | OSL1 | 2015 | 2017 | 2016 |
| OSL2 | 2015 | 2017 | 2016 | OSL2 | 2015 | 2017 | 2016 |
| OSL3 | 2015 | 2017 | 2016 | OSL3 | 2015 | 2017 | 2016 |
| SAN1 | 2017 | 2016 | 2015 | SAN1 | 2017 | 2015 | 2016 |
| SAN2 | 2017 | 2016 | 2015 | SAN2 | 2017 | 2015 | 2016 |
| SAN3 | 2017 | 2016 | 2015 | SAN3 | 2017 | 2015 | 2016 |
| SAN4 | 2017 | 2016 | 2015 | SAN4 | 2017 | 2015 | 2016 |
| TRD1 | 2017 | 2016 | 2015 | TRD1 | 2017 | 2016 | 2015 |
| TRD2 | 2017 | 2016 | 2015 | TRD2 | 2017 | 2016 | 2015 |
| TRD3 | 2017 | 2016 | 2015 | TRD3 | 2017 | 2016 | 2015 |
| TRD4 | 2017 | 2016 | 2015 | TRD4 | 2017 | 2016 | 2015 |

Model training
Hyperparameter tuning
independent model evaluation
not used

*Figure 1: Old setup vs new setup*

2)    Moreover, it would be better (but not strictly necessary, to my view) that the datasets are divided into four independent sets (i.e., the training, validation 1, validation 2 and test sets), as time lag selection also takes place according to the following lines: "Secondly, the structural parameters were fixed, and different lag values ranging from 1 hour to 200 hours were tested to identify the optimal lag value" (lines 219–220).

We agree with the reviewer that, having a fourth data set could improve the selection of the time lag values. However, we have decided, following the major comment of Reviewer#2, to redo the hyperparameter tuning using Bayesian optimization (Snoek et al., 2012) which is expected to improve the estimation of the hyperparameter values. The time lag will be considered as a tunable hyperparameter, following the study of Kratzert et al. (2019).

3)    In lines 217–216, it is written that "BERG1, OSL1, SAN1 and TRD1 roofs were selected to test different hyperparameters to find the optimal parameters for each city". Would it be better to select different hyperparameters for each roof?

We agree with the reviewer that it is better to tune hyperparameters for each roof individually. Accordingly, we will optimize the hyperparameters for each roof using Bayesian optimization.

4)    In lines 209–211, it written that "data were aggregated into one-hour resolution, and snow accumulation periods were excluded (1 Oct. – 31 Mar.). One year was used for training and one year for validation. The selection of the training year was based on the sum of precipitation as the wettest year between 2015 to 2017 for each roof, and the second wettest year for validation. The rationale for the

selection is that the wettest year covers a broader span of precipitation events which improves the generalization performance of the models". To my view, it would be better if the training and validation periods for all greens roofs were presented in a new table.

A table will be provided to present training, validation and testing periods for all the roofs

5) Also, I think that –at least in the supplement– it would be interesting to show what happens when one uses the entire datasets (i.e., without excluding the snow accumulation periods or other periods), and not selected parts of these datasets.

We initially used the entire data set for ML model training and validation. However, we have decided to remove snow periods to allow for comparison with the benchmark model (the conceptual retention model) which does not account for snow modelling.

6) I find that some important literature pieces on data-driven hydrological modelling (e.g., some of the oldest works in the field) are currently missing from the manuscript's reference list.

We are not aware about the literature pieces that is referred to by the reviewer. We attempted to present a literature review that balances between green roof modelling and the application of ML in hydrological modelling. We have mentioned some of the early work in ML modelling in hydrology e.g. (Daniell, 1991; Hsu et al., 1995; Karlsson & Yakowitz, 1987).

7) Lastly, since the manuscript is not typo-free at the moment, a careful reading and typo correction are required. For instance, something is currently wrong with the sections numbering ("2 Data", "2.1 Machine learning models", "3 Results and Discussion"). Also, there are typos in the units, symbols and equations, which should be written according to the following conventions:

• Single-letter variables should be written in italics.

• Multi-letter variables should not be written in italics.

We apologize for any typos in the manuscript and we will modify the errors identified by the reviewer.

**References**

Daniell. (1991). Neural networks. Applications in hydrology and water resources engineering. *National Conference Publication - Institution of Engineers, Australia*.

Hsu, Gupta, & Sorooshian. (1995). Artificial Neural Network Modeling of the Rainfall-Runoff Process. *Water Resources Research*. https://doi.org/10.1029/95WR01955

Karlsson, & Yakowitz. (1987). Nearest-neighbor methods for nonparametric rainfall-runoff forecasting. *Water Resources Research*, *23*(7), 1300–1308. https://doi.org/10.1029/WR023i007p01300

Kratzert, Klotz, Shalev, Klambauer, Hochreiter, & Nearing. (2019). Towards learning universal, regional, and local hydrological behaviors via machine learning applied to large-sample datasets. *Hydrology and Earth System Sciences*. https://doi.org/10.5194/hess-23-5089-2019

Snoek, Larochelle, & Adams. (2012). Practical Bayesian optimization of machine learning algorithms. *Advances in Neural Information Processing Systems*.

---

## Author Comment (AC2)

Dear Referee,

We would like to thank you for the thoughtful comments which will contribute towards improving the manuscript. The suggestion of automatically optimizing the hyperparameters of ML models has inspired us to navigate many optimization algorisms for hyperparameter tuning that are rarely applied in hydrological modelling studies.

*First, I want to apologize with Authors due to my late review. It was due to unexpected issues. The present study presents a numerical analysis to compare the performance of multiple Machine Learning techniques against conceptual models for the hydrological analysis and forecasting of Green Roofs behavior. The aim of the paper is interesting and of relevance for HESS readers.*

We appreciate the positive comment about the study.

*However, I find that the paper has multiple weaknesses:*

- *There are multiple bold statements against the use of physically-based models for GRs analysis, which are not supported by evidence and not needed in the manuscript, which should simply attain to its aim: assessing the performance of ML techniques for GRs analysis. Instead of reinforcing the paper, these statements draw the attention on other aspects, which are highly debatable. There doesn't exist a perfect numerical tools for everything, or one better than the other. It's up to the modeler to choose the right model for the specific modeling task.*

    We can agree that some statements in the current manuscript are debatable and not needed for the study. We will modify the manuscript accordingly.

- *The emulators training is performed by using the trial-and-error technique, which is an outdated and inefficient methodology. This is especially true for this task since the response surface in the hyperparameters' space can be multimodal, thus making it easy to get trapped in local minima. Furthermore, the uncertainty of the estimated hyperparameters should be properly assessed and eventually propagated in the validation step. The way it is handled in the paper (manually changing hyperparameters) is weak.*

    This is an excellent comment that has really inspired us to search for suitable algorithms for hyperparameter tuning. We acknowledge the limitation of trial-and-error techniques using random or grid sampling which we applied following recent hydrological modelling studies (King et al., 2020; Kratzert et al., 2018, 2019; Sattari et al., 2021; Teweldebrhan et al., 2020; Mo Zhang et al., 2020). Based on the reviewer comments, we decided to apply Bayesian optimization for hyperparameters tuning (Snoek et al., 2012) , which is suitable for functions in which evaluating one set of parameters is expensive and time-consuming. This algorithm was applied by Worland et al. (2018) to optimize hyperparameters for several machine learning algorithms to predict low flows for ungauged basins. In Bayesian optimization, the objective function (i.e. the relation between hyperparameters and the error value in the validation data set) is approximated by a probabilistic model (e.g.

Gaussian processes) that is used to select the most promising hyperparameter to evaluate in the true objective function.

- *Since authors calibrate (manually, but still calibrate) the emulators and compare it with a conceptual model, then the latter should be calibrated as well to conduct a fair comparison. This was not done.*

  The conceptual green roof retention model describes the processes that control moisture removal from the green roof substrate during dry weather periods. The model is implemented using a physically-based estimate of ET and a measurable physical property of the substrate, Smax. The parameter Smax represents the maximum retention capacity of the green roof or the difference between the field capacity and the permanent wilting point of the green roof substrate (Stovin et al., 2013). There exist standard laboratory tests to physically measure the substrate field capacity (FLL, 2008) and the permanent wilting point (Fassman & Simcock, 2012). It may be argued, therefore, that our conceptual model is physically based, and should not require calibration. No empirical parameters are introduced into the model that require calibration. However, we acknowledge that the Smax values were estimated in this case, and that it is relevant to confirm the extent to which calibration of this parameter would further improve the predictions. Therefore, we agree to calibrate the conceptual model (by tuning the Smax value) for each roof in the study. We will use the data of the training year for Smax tuning and the testing year for model evaluation and comparison with the ML models. As the conceptual model is cheap to run, we could run the conceptual model for all roofs by varying Smax between 1% to 100% of the roof substrate as shown in Figure 1.

[Figure]

*Figure 1: Tuning of Smax. vol is a measure of the volumetric error (equation 15 in the manuscript)*

**Specific Comments:**

L2-5 In my opinion, there is a general misunderstanding in this field, which is reiterated in multiple manuscripts, and it's the idea that conceptual models are always computationally cheaper than physically-based models for the hydrological analysis of GRs. Except particular circumstances, the computational cost is comparable. For instance, the authors can verify by themselves that HYDRUS-1D, a mechanistic hydrological model frequently used in GR analysis, takes less than few seconds for a long-term hydrological simulation. Conversely, for the same task, some conceptual models can be even more computationally expensive if the code is developed in excel or in high-level programming languages. Therefore, I would not build the premise of the work on this.

L2-5 Regarding the complexity, we should first define what is complexity (number of parameters, number of processes, etc). This is again questionable.

> As mentioned earlier, we agree to modify the manuscript by removing debatable statements about physically-based models which is not needed for the study.

Measurements: This is true and implies that conceptual models are not easily generalizable.

L20-25 "Improving quality" is a bold statement. There is an extensive literature about nutrients leaching from GRs.

>  This will be modified accordingly in the manuscript

L30 Why bold font?

>  We will remove the bold font

L35-40 I don't agree with these statements. Mechanistic models actually rely on huge literature body, which can be used to set the model parameters. For instance, parameters of the van Genuchten can be obtained with pedotransfer functions (using particle size distribution and other info from the producer) or set according to several studies which have been already performed. The unsaturated conductivity is needed as the soil water retention curve in the Richards equation, there is no difference. What is the acceptable level of uncertainty depends on the analysis (in dry conditions the magnitude of fluxes is low thus K is not prominent).

L55 Computational cost: As I stated before, I don't agree with this.

As mentioned earlier, we agree to modify the manuscript by removing debatable statements about physically-based models which is not needed for the study

L75-80 MLs are not uncertainty-free.

We agree. We intended to say that ML models reduce (not eliminate) the structural uncertainty that is caused by inadequate model assumptions (i.e. mathematical equations) in comparison to conceptual or physical models. We acknowledge that the performance of ML models is affected by the selection of suitable hyperparameters, which is a source of uncertainty. We will modify this part.

L115-120 "Green Roof runoff" should be "Green Roof subsurface runoff" to avoid misunderstandings.

>  We agree. This will be modified accordingly in the manuscript

I would just say " when observations are not available"

>  We agree. This will be modified accordingly in the manuscript

L168. "Trial-and-error" This is not true. A correct ANN training should use numerical optimization to identify the right set of hyperparameters since

>  As mentioned earlier, we will apply the Bayesian optimization for hyperparameter tuning

Section 2.2 I'm not sure that you can basically neglect physical properties of GRs. This might be somehow borderline acceptable for extensive GRs but morphological and hydraulic characteristic will play an important role as the soil substrate depth increases. This is acknowledged also in one of latest

paper from the same authors (Peng et al., 2020), and it is rather intuitive. I would be curious to see how the emulators behave when splitting the sample between thin and thick roofs. This would certainly deliver a more meaningful information to the community.

> This was done via transferring trained ML models between the green roofs. ML models that are trained using the data of thin roofs, even located in different locations, were used to simulate outflows of thick roofs and vice versa.

L210 The validation should be performed on a drier year to really assess the generalizability of emulators.

> We trained the ML models using the wettest years and was validated on the drier year

*L210-215 The optimal hyperparameters should be calibrated numerically, since you can easily end up trapped in a local minima (10.1016/j.jhydrol.2005.03.013). This is true for all emulators.*

*The use of Latin hypercube doesn't make solve the problem. You have a better coverage of parameters' space but, unless you use a global optimization strategy, you can be still trapped in local minima.*

> As mentioned earlier, we will apply the Bayesian optimization for hyperparameter tuning

L220 What are the structural parameters?

> This is meant to refer to the hyperparameters that describe the structure of the network (i.e. number of neuron and hidden layers) to differentiate them from other hyperparameters such the length of input sequence (lag).

L221 What you attempt to do is to investigate how small changes in hyperparameters affect the response of the emulator. Basically, how the uncertainty in the estimated hyperparmeters (you see that ML techniques are not uncertainty free) propagates. This should have been done more correctly by numerically optimizing MLs parameters and estimating (at least) their confidence intervals. Even better would have been using Bayesian inference to estimate posterior uncertainty (e.g., 10.1016/j.jhydrol.2011.09.002).

> As mentioned earlier, we will apply the Bayesian optimization for hyperparameter tuning.

L2.3 Why reporting all these equations, which are already mentioned in other studies from the same authors? Cite them and move forward.

> This will be done in the revised manuscript

L228 "Without the need of prior calibration…" This sounds puzzling to me. In the Introduction you write "calibration is needed to find their optimal values, unlike physically-based models", which is true since conceptual models generally needs site-specific calibration. If conceptual model parameters were not previously calibrated in other studies for the same site, then they should be calibrated here to conduct a fair comparison with trial-and-error optimized MLs.

> As mentioned earlier, we will calibrate the conceptual model by tuning the Smax value

L3.1 For the reasons that I mentioned above, I consider this way of training emulators not formally correct and scientifically outdated.

> As mentioned earlier, we will apply the Bayesian optimization for hyperparameter tuning.

L331 This can be said only when you perform a scientifically sounding calibration and uncertainty assessment of both models. None of the two was carried out, furthermore the conceptual model was not calibrated, thus the comparison is not fair.

> This will be achieved after optimizing hyperparameters of ML models using Bayesian optimization and calibrating the conceptual model by tuning the Smax value.

L333-335 Not sure what you refer with "…accommodate complex, multi-layered systems". These are bold statements not supported by evidence, which actually should be avoided since they don't contribute to the discussion unless they are proven.

> As mentioned earlier, we will remove any bold statements which are not important for the current study

**References**

Bengtsson, Grahn, & Olsson. (2005). Hydrological function of a thin extensive green roof in southern Sweden. *Nordic Hydrology*, *36*(3), 259–268. https://doi.org/10.2166/nh.2005.0019

Fassman, & Simcock. (2012). Moisture Measurements as Performance Criteria for Extensive Living Roof Substrates. *Journal of Environmental Engineering*. https://doi.org/10.1061/(asce)ee.1943-7870.0000532

FLL. (2008). Guidelines for the Planning , Construction and Maintenance of Green Roofing - Green Roofing Guideline.

(2020). Application of machine learning techniques for regional bias correction of snow water equivalent estimates in Ontario, Canada. *Hydrology and Earth System Sciences*. https://doi.org/10.5194/hess-24-4887-2020

Kratzert, Klotz, Brenner, Schulz, & Herrnegger. (2018). *Rainfall – runoff modelling using Long Short-Term Memory ( LSTM ) networks*. 6005–6022.

Kratzert, Klotz, Shalev, Klambauer, Hochreiter, & Nearing. (2019). Towards learning universal, regional, and local hydrological behaviors via machine learning applied to large-sample datasets. *Hydrology and Earth System Sciences*. https://doi.org/10.5194/hess-23-5089-2019

Sattari, Apaydin, Band, Mosavi, & Prasad. (2021). Comparative analysis of kernel-based versus ANN and

deep learning methods in monthly reference evapotranspiration estimation. *Hydrology and Earth System Sciences*. https://doi.org/10.5194/hess-25-603-2021

Snoek, Larochelle, & Adams. (2012). Practical Bayesian optimization of machine learning algorithms. *Advances in Neural Information Processing Systems*.

Soulis, Valiantzas, Ntoulas, Kargas, & Nektarios. (2017). Simulation of green roof runoff under different substrate depths and vegetation covers by coupling a simple conceptual and a physically based hydrological model. *Journal of Environmental Management*. https://doi.org/10.1016/j.jenvman.2017.06.012

Stovin, Poë, & Berretta. (2013). A modelling study of long term green roof retention performance. *Journal of Environmental Management*, *131*, 206–215. https://doi.org/10.1016/j.jenvman.2013.09.026

Teweldebrhan, Schuler, Burkhart, & Hjorth-Jensen. (2020). Coupled machine learning and the limits of acceptability approach applied in parameter identification for a distributed hydrological model. *Hydrology and Earth System Sciences*. https://doi.org/10.5194/hess-24-4641-2020

Worland, Farmer, & Kiang. (2018). Improving predictions of hydrological low-flow indices in ungaged basins using machine learning. *Environmental Modelling and Software*. https://doi.org/10.1016/j.envsoft.2017.12.021

Zhang, Mo, Shi, & Xu. (2020). Systematic comparison of five machine-learning models in classification and interpolation of soil particle size fractions using different transformed data. *Hydrology and Earth System Sciences*. https://doi.org/10.5194/hess-24-2505-2020

Zhang, Zheng, Szota, Fletcher, Williams, & Farrell. (2019). Green roof storage capacity can be more important than evapotranspiration for retention performance. *Journal of Environmental Management*. https://doi.org/10.1016/j.jenvman.2018.11.070

---

## Author Response (AR2)

In the manuscript, the authors evaluated the performance of 4 machine learning (ML) approaches in simulating runoff from green roofs. The experiments include 16 green roofs in 4 cities which are of different environmental conditions. Upon modeling retention of green roofs, The authors highlighted the advantage of the ML methods in comparing with a conceptual model. Upon modeling runoff, most models achieved a promising performance (NSE>0.5). The authors also examined the transferability of ML models between green roofs and concluded that those models could be transferred between cities with similar rainfall events characteristics.

As an extra reviewer who accidentally reviewed the out-of-date version, I am impressed by the substantial amount of work that the authors have done to improve the manuscript. The current version of the manuscript is clear-written and resolved most concerns I had in the previous version. I found several technical issues that I will detail following, but most of them are easy to fix in my opinion.

We appreciate the positive feedback of the reviewer. We acknowledge that the thoughtful comments of the reviewers in the first round have significantly contributed to improving the quality of the study. This indeed highlights the vital role of the peer-reviewing process.

L25: I like the highlight of retention and detention, as the first round manuscript did.

Done

L190: The author should write clearly about how the distance is calculated in this work, instead of "such as".

Done

L210: The description does not agree with the precipitation amount in table 2. For example, at Bergen and Oslo the drier year is chose as the training set.

The initial selection of the training periods was based on the amount of precipitation. However, after hyperparameter tuning, we further analyzed the change of ML performance when using the initially selected validations datasets for model training. Some of the validation datasets slightly improved the ML performance and hence were selected as training datasets. The final selection of the training, validation and testing periods is presented in table 2. This has been stated in the MS in lines 209-215.

L234: The dropout citation is not correct here as it is already so widely used before the cited work.

Done. We cited the original paper

L331: Table 5 presented the testing error?

Yes. We clarified that in the revised MS

L335: Typo. Reads BERG2 here but BERG1 in figure 6?

Done

L344: The author only explained why models on Bergen are of better performance comparing to Oslo, but did not reason upon "other roofs in the study" as claimed.

Corrected

L351 and figure 7: The author implied that the TRD site is heavily impacted by snow melting here, which is supported by the calibration results in table 4 (preferred longer lag time). However figure 7 did not present which season/month/date it plotted, and readers who are not familiar with the climate of Norway (like me) may have a hard time understanding the effect of snow storage.

Done.

L362: I am wondering if the conclusion is true for all sites? The three sites presented in figure 8 happen to have positive and negative biases for the two training years, and maybe that is why the sum of two models outplay either one of them. Are there any sites that the two models from two years result in biases of the same direction?

we found few green roofs where the two LSTM models from the two years resulted in biases of the same direction (at BERG3 and OSL3 roofs, as shown below). We clarified that in the revised MS lines 364-366

[Figure]

L383: In the manuscript, the author did not include a process-based or conceptual model upon detention process, and as a result, the work only proved that ML models are better at simulating retention process comparing to conventional methods, other than simulating the runoff. The reason seems to be the detention models are not "convincing" according to the manuscript. However, the authors reviewed detention models using three paragraphs (L39-57) in the introduction. Are none of those reviewed models "convincing"?

The reviewed detention models rely on calibration to estimate their parameter values. Previous studies have highlighted the limitation of transferring calibrated parameters between similar roofs

(Johannessen et al., 2019) and difficulties of identifying clear relationships between model parameters and roof characteristics  (Kasmin et al., 2010).  The ML methods have shown good potential in modelling detention (high NSE) and transferability between cities with similar climatic condition.

Table 1: There are two BERG2.

Corrected

Figure 4: I would suggest labeling train, validation, and test for each plot.

We prefer to keep the current labelling because the final selection of the training, validation and testing periods is different among the cities (table 2).

figure 6: I think the author presented the testing period other than the validation period? And which are the three months?

Corrected

In general, I find the manuscript is well-written and could be a novel contribution to the community. I have to acknowledge that I am not an expert in green roof modeling, and I may not fully understand the background and contribution of this manuscript, so please consider my review accordingly.

We thank the reviewer for the thoughtful comments and the positive feedback

**References**

Johannessen, Birgitte Gisvold, Hamouz, Vladimír, Gragne, Ashenafi Seifu, & Muthanna, Tone Merete. (2019). The transferability of SWMM model parameters between green roofs with similar build-up. *Journal of Hydrology*, *569*(October 2018), 816–828. https://doi.org/10.1016/j.jhydrol.2019.01.004

Kasmin, H., Stovin, V. R., & Hathway, E. A. (2010). Towards a generic rainfall-runoff model for green roofs. *Water Science and Technology*, *62*(4), 898–905. https://doi.org/10.2166/wst.2010.352